# Temporal analysis of T-cell receptor-imposed forces via quantitative single molecule FRET measurements

Janett Göhring [1,2,5], Florian Kellner [1,5], Lukas Schrangl [2,5], René Platzer[1], Enrico Klotzsch[2,3,4], Hannes Stockinger [1], Johannes B. Huppa [1✉] & Gerhard J. Schütz [2✉]

Mechanical forces acting on ligand-engaged T-cell receptors (TCRs) have previously been implicated in T-cell antigen recognition, yet their magnitude, spread, and temporal behavior are still poorly defined. We here report a FRET-based sensor equipped either with a TCR-reactive single chain antibody fragment or peptide-loaded MHC, the physiological TCR-ligand. The sensor was tethered to planar glass-supported lipid bilayers (SLBs) and informed most directly on the magnitude and kinetics of TCR-imposed forces at the single molecule level. When confronting T-cells with gel-phase SLBs we observed both prior and upon T-cell activation a single, well-resolvable force-peak of approximately 5 pN and force loading rates on the TCR of 1.5 pN per second. When facing fluid-phase SLBs instead, T-cells still exerted tensile forces yet of threefold reduced magnitude and only prior to but not upon activation.

[1] Institute for Hygiene and Applied Immunology, Center for Pathophysiology, Infectiology and Immunology, Medical University of Vienna, Vienna, Austria. [2] Institute of Applied Physics, TU Wien, Vienna, Austria. [3] Laboratory of Applied Mechanobiology, Department for Health Sciences and Technology, ETH Zürich, Zürich, Switzerland. [4] Institute of Biology, Experimental Biophysics/ Mechanobiology, Humboldt Universität zu Berlin, Berlin, Germany. [5] These authors contributed equally: Janett Göhring, Florian Kellner, Lukas Schrangl. ✉email: johannes.huppa@meduniwien.ac.at; schuetz@iap.tuwien.ac.at

A remarkable feature of the vertebrates' immune system is its inherent ability to distinguish harmful from harmless based on the primary antigen structure. T-cells embody this trait of adaptive immunity through their unique detection of antigens, which are displayed as peptide fragments in the context of the products of the major histocompatibility complex (MHC) on the surface of antigen-presenting cells (APCs). The formation of a productive immunological synapse, the area of contact between a T-cell and an APC[1], is driven by T-cell antigen receptors (TCRs) on the T-cell binding to peptide/MHC complexes (pMHC) on the APC. Importantly, T-cells specifically detect the presence of even a single antigenic pMHC molecule among millions of structurally similar yet non-stimulatory pMHCs[2]. This is even though nominal pMHC–TCR interactions are of a rather moderate affinity, at least when measured in vitro. Despite considerable interest, the molecular, biophysical, and cellular mechanisms underlying this phenomenal sensitivity and specificity are not well understood.

Tensile forces have been implicated in the discrimination of antigenic peptides from non-activating pMHCs, especially if T-cells gauge TCR-proximal signaling based on TCR–pMHC bond lifetimes as is proposed by the kinetic proofreading model[3–6]. Over the course of the last decade such forces have garnered significant attention and have even been regarded by some an instrumental trigger in T-cell antigen detection[7,8]. With the use of a single-molecule Förster resonance energy transfer (FRET)-based live-cell-imaging assay, we have observed synaptic unbinding between TCR and pMHC with significantly increased off-rates compared to TCR–pMHC binding in solution[9], possibly as the result of cellular motility. In addition, cell mechanical[10,11] and in vitro[12] studies have revealed altered TCR–pMHC unbinding under force, yielding indications for the relevance of both catch bonds[10], as well as slip bonds[12] for the antigen discrimination process.

T-cells impose forces onto the TCR through different cytoskeletal elements. Upon activation, TCR microclusters are migrating towards the center of the immunological synapse by means of actin flow[13] and coupling to the microtubule motor dynein[14]. The results of a more recent study suggest that, single TCRs are directly transported upon triggering via single motor protein molecules along the actin cytoskeleton[15].

When confronted with beads, which have been functionalized with TCR-ligands, T-cells undergo calcium signaling upon mechanical bead retraction[7,16]. Activation thresholds were found markedly reduced when pMHC-coated beads were moved tangentially to the T-cell surface[15]. In line with this, T-cells were reported to push and pull against model APCs in the course of antigen-driven activation[17,18]. It has also been proposed that T-cells actively probe the elastic properties of the substrate[19], and global pulling forces of up to 300 pN were observed when T-cells had been activated via antigen-functionalized elastomeric pillar arrays[20] or polymers with embedded beads[21,22].

These and related findings prompted efforts to quantify mechanical forces directly within the immunological synapse. Experiments involving the use of DNA-based tension sensors have given rise to the suggestion that forces larger than 12 pN were acting on ligand-engaged TCRs when T-cells were confronted with antigenic glass-immobilized pMHCs[23]. Forces amounting to up to 4.7 pN were reported for T-cells that interacted with planar glass-supported lipid bilayers (SLBs) featuring pMHCs anchored to gold particles[24]. Furthermore, when an adjustable upper force limit was imposed on TCR–pMHC bonds with the use of rupturable DNA tension gauge, the recruitment of the TCR-proximal tyrosine kinase ZAP70 to the plasma membrane, a process that is integral to early T-cell signaling, correlated well with the maximal amplitude of the allowed force[23].

However, current ensemble measurements render it challenging to assess pulling forces acting on individual TCRs. Typically, multiple bonds are simultaneously subjected to strain, and it is not clear whether the DNA-based tension sensors are arranged in parallel or zipper-like configuration[25]. Furthermore, when considering the mechanism underlying the function of such sensors, there is a strong dependence of the unzipping force on the loading rate[26] as well as on the duration of the applied force[27], which constitute a priori unknown parameters. If given sufficient time, all DNA duplexes will eventually separate when any force is applied. Last but not least, ensemble-based readout modalities fail to afford kinetic force profiles acting on individual receptor–ligand bonds which hence remain undetermined.

In this study, we report the engineering and application of a quantitative FRET-based force sensor for application within the immunological synapse, which operates at the single-molecule level. As a spring element, we employed a peptide derived from the spider silk protein flagelliform comprised of 25 amino acids with known elastic properties[28,29]. The spring peptide was anchored to either gel-phase or fluid-phase SLBs and conjugated either to a single-chain antibody fragment (scF$_V$) derived from the TCRβ-reactive H57 monoclonal antibody (H57-scF$_V$) to serve as a high-affinity artificial TCR-ligand, or to MHC loaded with a nominal peptide as the physiological TCR-ligand. In its collapsed configuration the sensor spans a distance of <8 nm and is therefore compatible with length restrictions of ~14 nm as they apply within the immunological synapse[30,31]. When engaging TCRs with the high-affinity ligand H57-scF$_V$, we observed 5–8 pN average forces per TCR for T-cells contacting gel-phase SLBs. Compared to H57-scF$_V$, force readings obtained with the use of the natural ligand pMHC showed a broader distribution shifted towards lower forces amounting to about 2 pN. Of note, only barely measurable TCR-imposed forces below 2 pN were detected when T-cells interacted with fluid-phase SLBs. Since both gel-phase and fluid-phase SLBs support T-cell stimulation, our measurements imply that mean perpendicular tensile forces smaller than 2 pN are already effective during the initiation of TCR signaling. Our findings render it furthermore highly unlikely that forces, at least to an extent they are measurable with our FRET-based system, are required for continual TCR-signaling at later stages of synapse formation.

## Results

**Design of the force sensor**. We designed the molecular force sensor (MFS) in a fashion that rendered it compatible with the use of protein-functionalized SLBs. The latter has been widely employed as surrogates of APCs[1], especially with the incorporation of nickel-chelating lipids, such as 1,2-dioleoyl-sn-glycero-3-[(N-(5-amino-1-carboxypentyl)iminodiacetic acid)succinyl] (Ni$^{2+}$) (Ni-NTA-DGS), which supports decoration of SLBs with poly-histidine (His)-tagged proteins of choice in experimentally definable densities[9]. The use of SLBs also allows for the application of total internal reflection-(TIR) based imaging modalities, which give rise to sufficiently low background noise for recording single-molecule traces.

Monovalent recombinant streptavidin (mSAv) was utilized in our study as an anchor unit for the force sensor to be embedded in the SLB (Fig. 1a, b). To ensure stoichiometric anchorage we engineered the tetrameric mSAv complex to contain one "alive", i.e. biotin-binding subunit and three "dead" subunits, which had been mutated to no longer bind biotin. Instead, dead subunits featured a tag comprised of six histidine residues each (mSAv-3xHis$_6$, for more details refer to "Methods" section) for attachment to SLBs employed in our study, which contained 2% Ni-NTA-DGS for functionalization with poly-histidine-

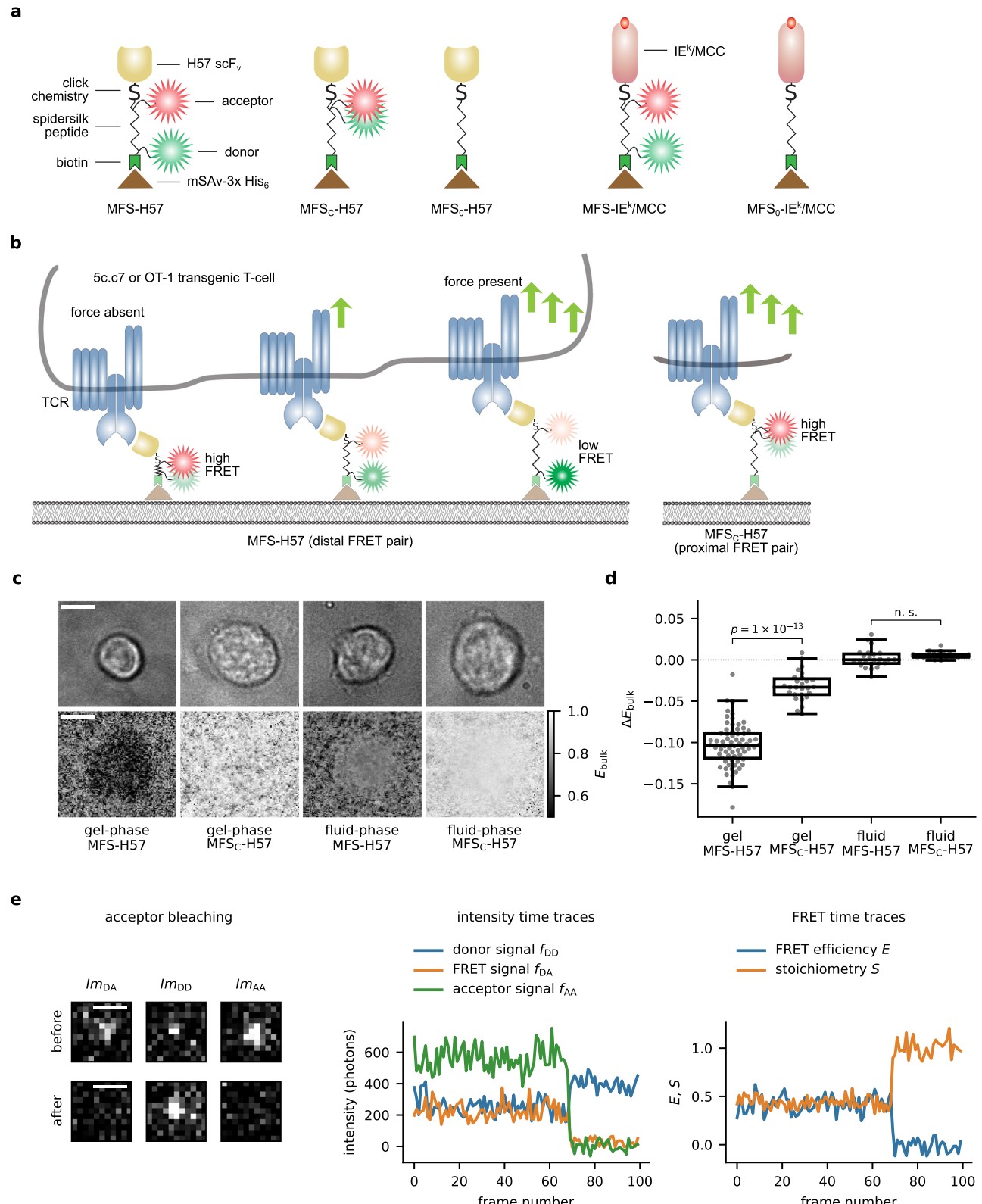

tagged proteins. Depending on the experiment, the carrier lipids (98%) were either 1,2-dipalmitoyl-sn-glycero-3-phosphocholine (DPPC) giving rise to immobile, gel-like SLBs which allowed for shear and pulling forces to take effect, or 1-palmitoyl-2-oleoyl-glycero-3-phosphocholine (POPC), which enabled high lateral protein mobility, and were as a consequence permissive for pulling forces only.

We designed the MFS to support direct visualization and quantification of single-digit piconewton forces via FRET with a temporal resolution in the millisecond range, as they were expected to occur between receptor–ligand pairs within the immunological synapse (Fig. 1a). Integral to the MFS were two corresponding fluorophores acting as a FRET pair and flanking the sensor's core which was derived from the spider silk protein

**Fig. 1 FRET-based sensor designed to detect forces on a single-molecule level. a** Schematic illustration of the force sensor constructs MFS-H57, and MFS-IE$^k$/MCC, the control construct MFS$_C$-H57, and the unlabeled constructs MFS$_0$-H57 and MFS$_0$-IE$^k$/MCC. **b** Exertion of forces stretches the spring core module, which reduces FRET yields for the molecular force sensor (here: MFS-H57) but not for the control force sensor (here: MFS$_C$-H57). **c** Representative images illustrating FRET efficiencies determined on a pixel-by-pixel basis via DRAAP on gel-phase (DPPC) and fluid-phase (POPC) SLBs with the use of the MFS or the MFS$_C$ construct (bottom row), and the corresponding transmission light microcopy images (top row). **d** Statistical analysis of ensemble FRET measurements. Each data point corresponds to a single DRAAP experiment of up to three cells in the field of view; boxes indicate the interquartile range (first quartile to third quartile), whiskers extend 1.5 times the interquartile range from the first and third quartile, the line indicates the median. A significant reduction in FRET efficiency ($\Delta E_{bulk}$) was only observed for MFS on gel-phase SLBs (n.s., not significant; two-sample Mann–Whitney $U$ test). $n = 68$ cells for MFS on gel-phase SLBs, $n = 24$ for MFS on fluid-phase SLBs, $n = 27$ for MFS$_C$ on gel-phase SLBs, $n = 21$ for MFS$_C$ on fluid-phase SLBs. **e** Representative single-molecule signals and time traces. The left panel depicts the fluorescence signals prior to and after photobleaching of the FRET acceptor. The middle panel shows single-molecule brightness values of the FRET donor, the FRET signal itself and the FRET acceptor. Bleaching of the acceptor could be observed as a correlated step-wise decrease of FRET and FRET acceptor intensities and an increase in FRET donor intensity. The right panel shows the stoichiometry and FRET efficiency. Scale bars: 5 μm (panel **c**), 1 μm (panel **e**).

flagelliform and consisted of five repeats of the peptide motif GPGGA[28,32]. We expected the spring to collapse with high FRET yields associated with the absence of force, yet upon force application to give rise to reduced FRET yields due to an increased distance separating the two fluorophores[29].

For a quantitative readout, the core module of the MFS had to (i) be amenable to site-specific modification with fluorophores of choice, (ii) afford directional conjugation of the spring module to the T-cell ligand, and (iii) provide anchorage to SLBs (Fig. 1a, b). To meet these criteria, we synthesized the spring peptide with an unpaired cysteine and a single lysine residue for fluorophore conjugation via maleimide and succinimidyl chemistry, respectively. The peptide's N-terminus was biotinylated to omit the N-terminal amino group and to confer binding of the peptide to SLB-resident mSAv-3xHis$_6$[33]. C-terminal incorporation of an ε-N$_3$-modified lysine residue in the sensor's core facilitated covalent attachment of site-specifically dibenzylcyclooctyne- (DBCO-) conjugated TCR–ligand via biorthogonal copper-free click chemistry[34].

As a TCR-engager we generated two constructs: first, we utilized the TCRβ-reactive recombinant H57-scF$_V$ (MFS-H57; for construction see Supplementary Fig. 1a–d), as we anticipated this to give rise to stable TCR-bonds, which unlike pMHC–TCR interactions sustain the tensile forces applied in the synapse without rupture. It is known that monomeric membrane-bound H57 activates T-cells with similar activation thresholds as agonistic pMHC[35,36]. Second, we also functionalized the force sensor with MHC class II molecule IE$^k$ loaded with a moth cytochrome c (MCC, IE$^k$/MCC) peptide (MFS-IE$^k$/MCC; for construction see Supplementary Fig. 1A–D), the nominal and physiological antigen of 5c.c7 TCR-transgenic CD4+ T-cells, which we employed predominantly in this study (see below). Importantly, the N-terminal biotin residue within the MFS core allowed for stoichiometric coupling to mSAv-3xHis$_6$ present on the SLB (Supplementary Fig. 1E).

In some experimental settings we set out to measure forces exerted by individual TCRs under moderate levels of antigenic stimuli (see below). To this end we diluted the fluorophore signals with an unlabeled force sensor termed MFS$_0$-H57 or MFS$_0$-IE$^k$/MCC. In addition, a stretch-insensitive version of the sensor was constructed to serve as a control (MFS$_C$-H57). Here the FRET acceptor and FRET donor fluorophores were separated by only a single amino acid (Fig. 1a, b).

Integrity of all employed constructs was confirmed by SDS–PAGE (Supplementary Fig. 1E). When anchored to SLBs, all constructs gave rise to robust T-cell activation as monitored via the rise in intracellular calcium (Supplementary Fig. 1F–G).

**Forces in the immunological synapse visualized at the ensemble level.** CD4$^+$ helper T-cells were isolated prior to their experimental use from the spleen and lymph nodes of 5c.c7 TCR transgenic mice and stimulated ex vivo for one week with antigenic peptide to promote proliferation and differentiation into antigen-experienced T-cell blasts. These were then seeded onto SLBs functionalized with intercellular adhesion molecule 1 (ICAM-1), the co-stimulatory molecule B7-1, and MFS-H57 or MFS$_C$-H57 (50–100 molecules per μm$^2$). We first confronted T-cells with gel-phase SLBs prepared from DPPC, with associated proteins featuring minimal diffusion ($D \leq 1.5 \times 10^{-4}$ μm$^2$ s$^{-1}$; Supplementary Fig. 2A). As shown by calcium imaging (Supplementary Figs. 1F and 3A) and the analysis of immune synapse recruitment of ZAP70 (Supplementary Fig. 3D), T-cells responded to such SLBs in a highly sensitive manner. Peak calcium signals were identical with those of the positive control, which involved stimulation via SLB-anchored IE$^k$/MCC, and elevated calcium flux was sustained for at least 15 min. We noticed a marginally delayed onset of T-cell activation triggered by force sensor-constructs when compared to that of the positive control, possibly reflecting slightly altered stimulation conditions within the synaptic space constraints[37]. Furthermore, the high viscosity inherent to DPPC-based SLBs prevented clustering of TCR-engaged SLB-resident MFS-H57 (Supplementary Fig. 2B), which we have previously observed to affect antibody-mediated T-cell activation[36].

Ensemble FRET efficiencies ($E_{bulk}$) were quantified via donor recovery after acceptor photobleaching (DRAAP) by comparing the average donor signal in a chosen region of interest before and after the photoablation of the acceptor fluorophore. A substantial reduction in $E_{bulk}$ underneath the T-cell indicated the elongation of the force sensor (Fig. 1c) which we did not observe when applying the control construct. For the statistical analysis, we calculated the difference $\Delta E_{bulk}$ by subtracting the region-matched $E_{bulk}$ recorded on a functionalized SLB in the absence of T-cells from $E_{bulk}$ underneath the T-cell, yielding a reduction $\Delta E_{bulk}$ of $-0.104$ (median). In contrast, the control construct showed a significantly smaller FRET-reduction $\Delta E_{bulk}$ of $-0.033$ (Fig. 1d).

To account for the comparably high mobility of MHC, ICAM-1, and B7-1 present on the surface of living APCs[38], we confronted T-cells with fluid-phase SLBs which were prepared from POPC and subsequently functionalized with proteins in the same fashion as the gel-phase SLBs. SLB-anchored MFS-H57 that underwent lateral diffusion with $D$ equaling $0.72 \pm 0.02$ μm$^2$ s$^{-1}$ (Supplementary Fig. 2A), were rapidly recruited into TCR-microclusters, transported to the center of the synapse (Supplementary Fig. 2B) and resulted in a robust calcium response and ZAP70 recruitment (Supplementary Figs. 1F and 3B, D), albeit with a slightly delayed onset of the calcium response. However, we did not observe significant reductions in FRET within synaptic MFS-H57 ($\Delta E_{bulk} = 0.000$ median for the MFS-H57 and $\Delta E_{bulk} = 0.005$ for the MFS$_C$-H57) (Fig. 1d).

Together, ensemble-based FRET experiments indicate the presence of tensile forces exerted by T-cells via their TCRs. FRET-changes were, however, only detectable when T-cells interacted with gel-phase but not fluid-phase membranes. Our observations suggest that TCR-exerted force fields acted foremost tangentially to the T-cell surface: in such cases, gel-phase membranes provided adequate counterforces to stretch the MFS-H57, while fluid-phase membranes would allow for MFS-H57 dragging without any recordable changes in FRET.

**Forces within the immunological synapse at the single molecule level.** While informative on a qualitative level, ensemble FRET measurements via the MFS are only of limited use for quantitative analysis for a number of reasons. First and foremost, any information regarding the kinetics and amplitude of single-molecular forces is masked in the ensemble average, which includes in addition to sensors under tension also contributions from unbound force sensors or sensors bound to TCR-enriched microvesicles[39], rendering the quantification of individual TCR-exerted forces impossible. Moreover, defective sensors may contribute to observed ensemble FRET efficiencies in ways that are difficult to account for or predict. Last, but not least, molecular fluorophore enrichment within the synapse in the course of T-cell activation could affect the FRET-readout.

To circumvent these limitations altogether, we carried out single-molecule microscopy experiments, which allowed us to filter out defective sensors and quantify forces between individual TCRs and SLB-anchored MFS-H57 in time and space with the use of both gel-phase and fluid-phase SLBs for T-cell stimulation. For this, T-cells were seeded onto SLBs functionalized with ICAM-1, B7-1, the unlabeled sensor $MFS_0$-H57 (50–100 $\mu m^{-2}$), and low concentrations of the MFS-H57 or $MFS_C$-H57 (<0.01 $\mu m^{-2}$). As shown in Supplementary Fig. 3, the stimulatory potency of such SLBs was verified by monitoring the calcium response and ZAP70-recruitment of interacting T-cells.

After low enough MFS-H57 densities were chosen to allow for the emergence of well-isolated single-molecule signals within the immunological synapse, we imaged the donor and acceptor channel at alternating excitation over time and calculated the FRET efficiency $E$ via sensitized emission, taking into account the corresponding cross-talk signals (see the section "Methods" for a detailed description and Fig. 1e for exemplary traces)[40]. We also determined the stoichiometry parameter $S$, which reflects the fraction of donor molecules in donor–acceptor complexes[40]. When plotting $S$ against $E$ for each observed data point, we found a single population centered around $S \sim 0.5$, testifying to the robustness of the filtering steps for single and functionally intact MFS-H57 (Supplementary Fig. 4; see Supplementary Fig. 5 for more details on the procedure). In the absence of T-cells we observed for MFS-H57 and $MFS_C$-H57 pronounced maxima at $E = 0.87$ and $E = 1.00$, respectively, which represented the FRET efficiencies of collapsed springs. The higher FRET efficiency recorded for the $MFS_C$-H57 construct resulted in all likelihood from the shorter distance within the FRET donor–FRET acceptor dye pair, regardless of the spring extension.

Inspection of the immunological synapse revealed considerable differences in the spectral behavior of the two MFS constructs: while $MFS_C$-H57 showed FRET efficiencies that were almost identical to those recorded in the absence of cells ($E = 0.99$), for MFS-H57 we detected a second peak at $E = 0.42$ with a weight of 21% in addition to the high-FRET peak at $E = 0.87$. This low-FRET peak corresponded to MFS-H57 subjected to force-induced elongation via ligated TCRs.

To read out the applied tensile forces from determined FRET efficiencies we factored in the elastic properties of the flagelliform

force sensor, which were previously determined in combined single molecule force spectroscopy and FRET experiments on a similar construct[29]. We inferred the relationship between FRET efficiency and the tensile force acting on the sensor used in our study (see the "Methods" section for a detailed description of the calibration) and calculated an average pulling force $F$ of 4.9 ± 0.3 pN for the low FRET peak at $E$ equaling 0.42 ± 0.03 (Fig. 2a, first panel). Expectedly, pulling forces were abrogated when inhibiting actin polymerization in the presence of cytochalasin D (Supplementary Fig. 6A).

To assess the dynamic range of our sensor, we estimated the minimum and maximum FRET efficiency of the low FRET peak that could be measured with the use of the MFS-H57. The smallest detectable FRET value $E_{min}$ was set by the donor–acceptor distance at maximum load, the Förster radius of the chosen FRET pair, and the signal to noise ratio of the data. As is laid out in more detail in the "Methods" section, we estimated $E_{min}$ to amount to 0.11, which relates to a tensile force $F_{max}$ of 10 pN. Furthermore, the largest detectable FRET value $E_{max}$ was limited by the overlap with the high FRET peak at $E = 0.87$. With the observed proportions of the two peaks we estimated $E_{max}$ to ~0.8, which related to a minimum detectable force $F_{min}$ of about 0.9 pN. We therefore conclude that the observed force peaks fell well within the sensor's dynamic range, and that they corresponded to a not fully extended spring element.

In line with the results of the ensemble measurements (see above) we did not observe a high force peak for sensors within synapses when applying fluid-phase SLBs prepared from POPC (Fig. 2a, second panel, Supplementary Fig. 7, and Supplementary Table 1). MFS-H57 binding to the TCR was confirmed in single-molecule tracking experiments in which we found the majority of MFS-H57 (58%) to exhibit a substantially reduced mobility underneath T-cells. In control experiments involving human Jurkat T-cells, which do not engage the MFS-H57 for lack of H57-$scF_V$-reactivity, we did not observe synaptic MFS-H57 immobilization (Supplementary Fig. 8).

**TCR-imposed forces under scanning conditions.** Next, we aimed to assess the degree to which the high number of molecular bonds formed between the SLB and the T-cell disperses cellular forces over multiple TCR–ligand bonds while lowering the forces acting on individual MFS molecules and their associated TCRs. Of note, such parallel arrangement of multiple TCR–ligand bonds may very well apply to activated T-cells featuring a fully established synapse, yet to a much lesser extent to T-cells scanning APC surfaces for antigenic ligands, where isolated TCR–ligand interactions become likely subject to larger tensile forces. To discriminate between these scenarios, we seeded T-cells onto ICAM-1-functionalized SLBs featuring MFS-H57 in low densities below the activation threshold (<0.01 molecules per $\mu m^2$) (Supplementary Fig. 3A, B, and D). Again, we detected a distinct high force peak corresponding to 6.0 ± 0.6 pN for T-cells confronted with gel-phase SLBs, which slightly exceeded the force peak measured for fully activated T-cells (Fig. 2a, third panel). We noticed a shoulder towards lower applied forces corresponding on average to 1.9 ± 0.2 pN for T-cells contacting fluid-phase SLBs (Fig. 2a, fourth panel). Forces of such little magnitude may reflect drag perpendicular to the T cell surface. However, since membrane alignment may be incomplete while T-cells scan the SLB for antigen, forces that end up stretching the sensor may also act tangentially with regard to the T-cell surface.

Next, we took advantage of the higher mobility of unbound sensors when compared to that of bound sensors, and specifically analyzed the force histograms for bound and unbound MFS-H57

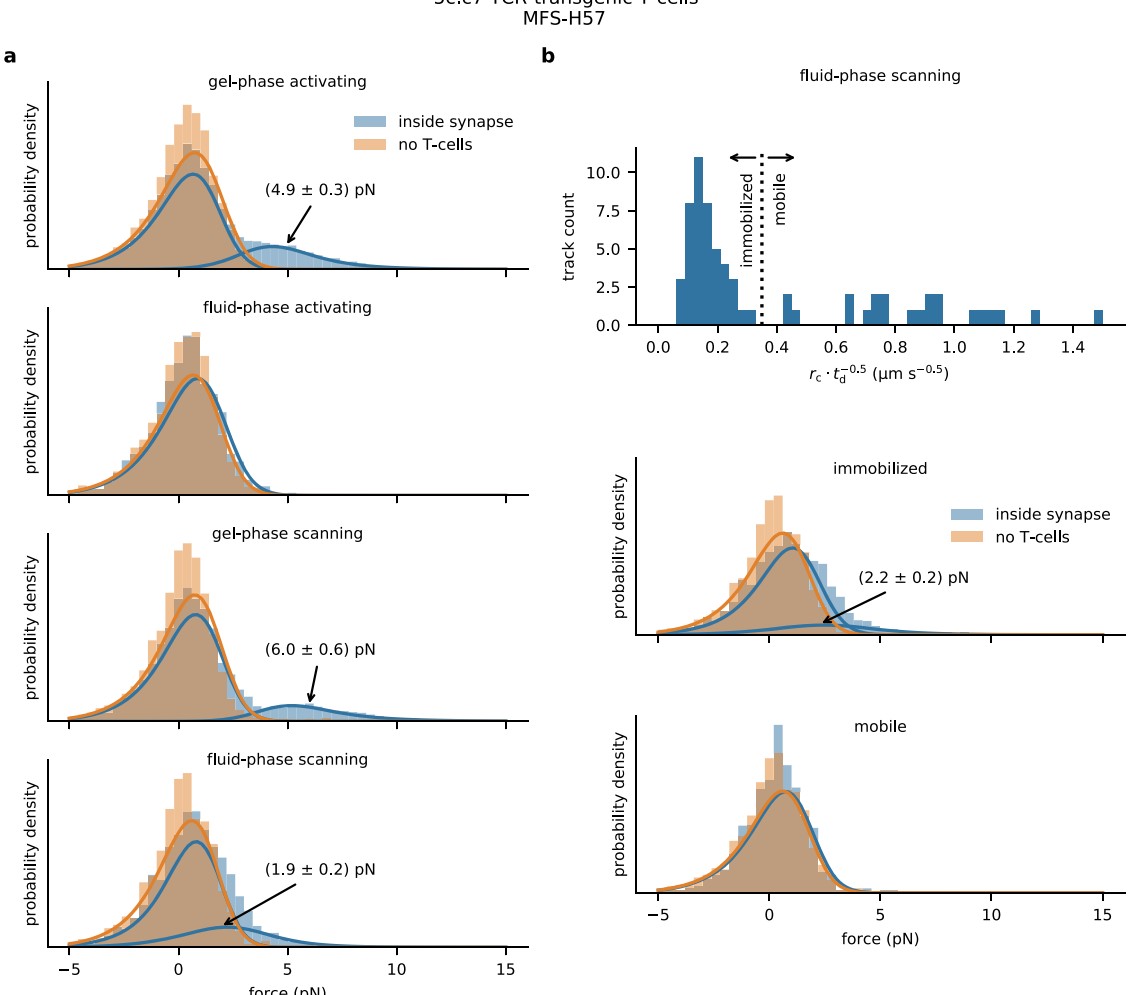

**Fig. 2 Single-molecule pulling forces exerted via individual TCRs on the H57-functionalized molecular force sensor. a** Single-molecule TCR-imposed forces determined within synapses of activated or scanning T-cells confronted with gel-phase (DPPC) or fluid-phase (POPC) SLBs. Experiments were performed on 5c.c7 T-cells using sensors functionalized with H57. A high force peak emerged for activated or scanning T-cells confronted with gel-phase SLBs at $F = 4.9 \pm 0.3$ pN and $F = 6.0 \pm 0.6$ pN, which corresponded to 21% and 15% of observed events, respectively. For T-cells scanning fluid-phase SLBs we observed an additional high force peak at $F = 1.9 \pm 0.2$ pN with 25% weight. $n = 15{,}556$ (593 movies, each containing $\geq 1$ cell) for MFS on gel-phase SLBs in contact with activated T-cells, $n = 12{,}657$ (195 movies) for MFS on fully stimulatory gel-phase without T-cells, $n = 6006$ (453 movies) for MFS on gel-phase SLBs including scanning T-cells, $n = 5502$ (165 movies) for MFS on gel-phase SLBs (non-stimulatory) without T-cells, $n = 1161$ (293 movies) for MFS on fluid-phase interacting with activated T-cells, $n = 1528$ (90 movies) for MFS on stimulatory fluid-phase without T-cells, n = 2147 (333 movies) for MFS on fluid-phase SLBs in contact with scanning T-cells, $n = 1796$ (70 movies) for MFS on non-stimulatory fluid-phase without T-cells. **b** Bound MFS were distinguished from unbound MFS on fluid-phase SLBs under scanning conditions by means of the single molecule mobility as described in Supplementary Fig. 8. The dashed line indicates the chosen boundary drawn between mobile and immobilized MFS. Unbound MFS (termed "mobile") gave rise to a force distribution showing no significant difference from the control scenario recorded without T-cells. Bound MFS (termed "immobilized") featured a high-force shoulder similar to Fig. 2a, fourth panel.

(Fig. 2b). As expected, unbound, i.e. mobile MFS-H57 did not register any tensile forces, while immobilized, i.e. TCR-engaged sensors gave rise to a shoulder in the force peak corresponding to $2.2 \pm 0.2$ pN, a value that was found in agreement with the global distribution of the combined data and about three-fold below that measured for TCRs interacting with immobile MFS-H57. Together, the data acquired at low MFS-H57 concentrations imply that perpendicular pulling forces exerted via the TCR are not a consequence of T-cell activation and may well precede the actual activation step.

**Spatiotemporal analysis of TCR-imposed forces**. An interesting aspect of our observations on gel-phase SLBs relates to the similarity between tensile forces applied by T-cells scanning for antigenic peptide and those imposed by activated T-cells. To further scrutinize the obtained data, we pooled the data in windows of 5 min after initial T-cell–SLB contact. The results revealed remarkable differences between scanning and activated T-cells. For activated T-cells (Fig. 3a), we observed the emergence of the high force peak only after ~10 min at $5.6 \pm 0.1$ pN, where it remained afterward. Tensile forces occurred both in the cell periphery as well as in central regions of the immunological synapse (Supplementary Fig. 9A). Quite in contrast, scanning T-cells exhibited the high force peak immediately following initial SLB contact (Fig. 3b). Interestingly, the average TCR-imposed pulling force declined over time from $7.5 \pm 0.2$ pN within the first 5 min to $6.3 \pm 0.2$ pN 5–10 min after contact, and later down to

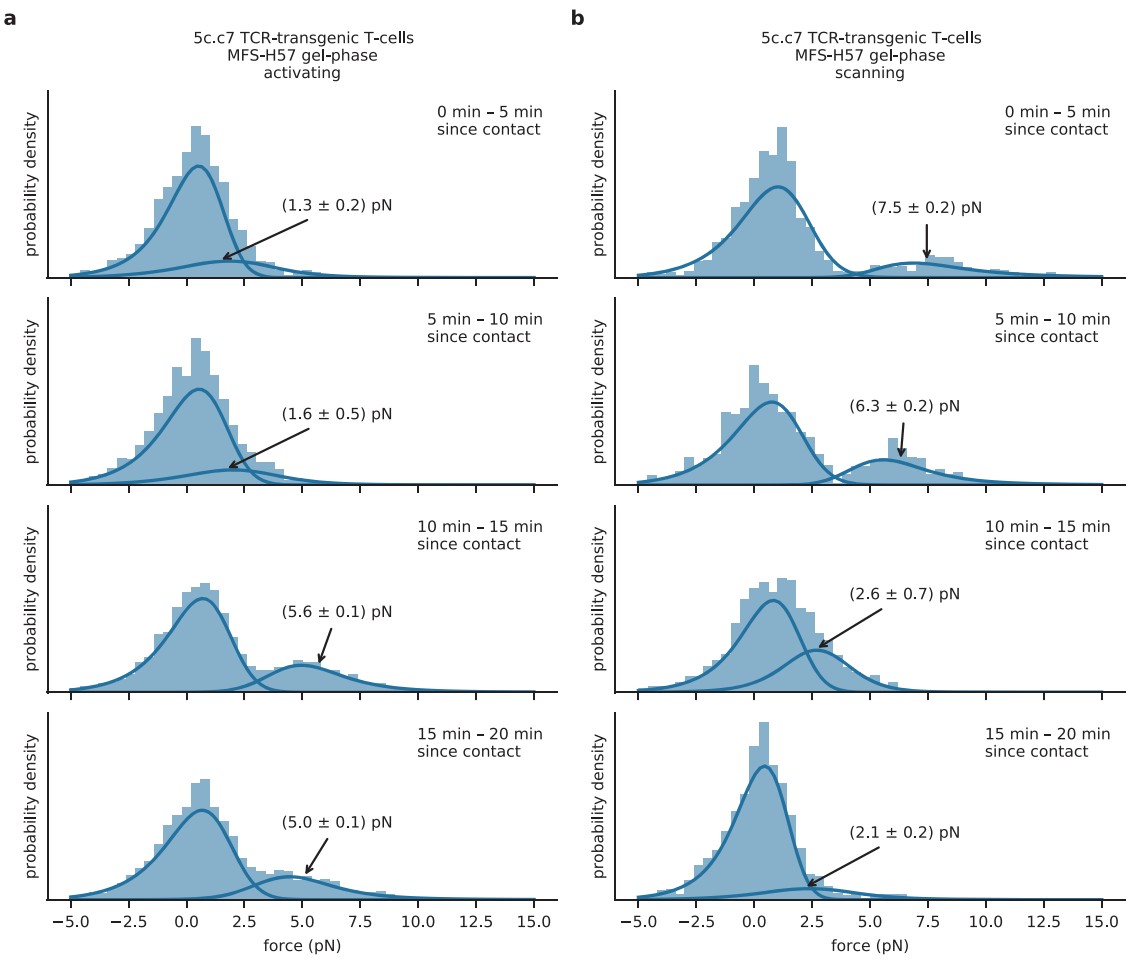

**Fig. 3 Single-molecule force profiles in the course of synapse formation.** We used MFS-H57 data recorded on 5c.c7 T-cells contacting activating (**a**) or non-activating (**b**) gel-phase SLBs. Data were grouped with respect to the time elapsed since the corresponding T-cell first contacted the SLB. Fit results for the average pulling forces are indicated in the figure. The following number of data points were used. Activating conditions: 0–5 min, $n = 1579$ (38 movies); 5–10 min, $n = 2701$ (60 movies); 10–15 min, $n = 3403$ (69 movies); 15–20 min, $n = 4047$ (75 movies). Scanning conditions: 0–5 min, $n = 1140$ (29 movies); 5–10 min, $n = 927$ (21 movies); 10–15 min, $n = 1105$ (32 movies); 15–20 min, $n = 1410$ (32 movies).

low force magnitudes below 3 pN. Again, we did not observe a spatial preference of forces (Supplementary Fig. 9B).

**TCR-imposed forces exerted by cytolytic CD8 + OT-1 transgenic T-cells.** While antigen recognition by CD4+ and CD8+ T-cells shares many underlying molecular and subcellular processes, functional consequences differ considerably. Since many studies on TCR-imposed forces have been conducted with the use of CD8+ T-cells[15,23], we also quantitated tensile forces with the use of cytolytic T-cells (CTLs) isolated from spleen and lymph nodes of OT-1 TCR-transgenic mice and differentiated ex vivo. As shown in Supplementary Fig. 6B, force analysis involving the use OT-1 TCR-transgenic CTLs interacting with MFS-H57 presented on gel-phase SLBs gave rise to histograms that were similar to those observed earlier with the use of CD4+ 5c.c7 T-cells. They yielded a high force peak of 6.1 ± 0.1 pN under activating and 5.2 ± 0.1 pN under scanning conditions.

**Temporal analysis of forces acting on single TCR-bound MFS molecules.** We next determined the force kinetics of single TCR-bound MFS-H57 molecules. We exclusively analyzed MFS-H57 entities which could be ascribed without ambiguity to the low-FRET peak, i.e. trajectories with at least five FRET efficiency data points below 0.6. The variability in the FRET signals within single traces was unexpectedly low: we observed an average standard

deviation per trajectory of 0.12 (corresponding to force fluctuations of 1.5 pN), which was substantially lower than the standard deviation derived from all low-FRET signals (equaling 0.22 and corresponding to 2.8 pN). This indicates that MFS-exerted forces covered a broad range but remained rather constant over the entire duration of the trajectories (on average 0.9 s recorded at 50 frames per second). To investigate transitions in force amplitudes applied on individual TCRs in more detail, we conducted experiments at considerably lower frame rates with trajectories recorded for up to 71 s on average. With decreasing frame rates, we observed more frequent force transitions within single trajectories. Accordingly, the dispersion of force values per single-molecule trajectory increased towards the dispersion of all recorded force data. Based on this and the representative single-molecule FRET trajectories shown in Fig. 4, we conclude that transitions between FRET levels occurred on a time scale of seconds.

For global analysis, we pooled all clearly identifiable force ramps from several experiments by synchronizing the traces with respect to the force peak, yielding an average single-molecule pulling trace (Fig. 4c). The force increased with an average loading rate of 1.5 pN s$^{-1}$ for ~2.5 s, before the accumulated force was released in a step-wise fashion. This behavior was reminiscent of load-fail events observed in traction-force microscopy[21], yet at 100-fold smaller peak force levels, and could

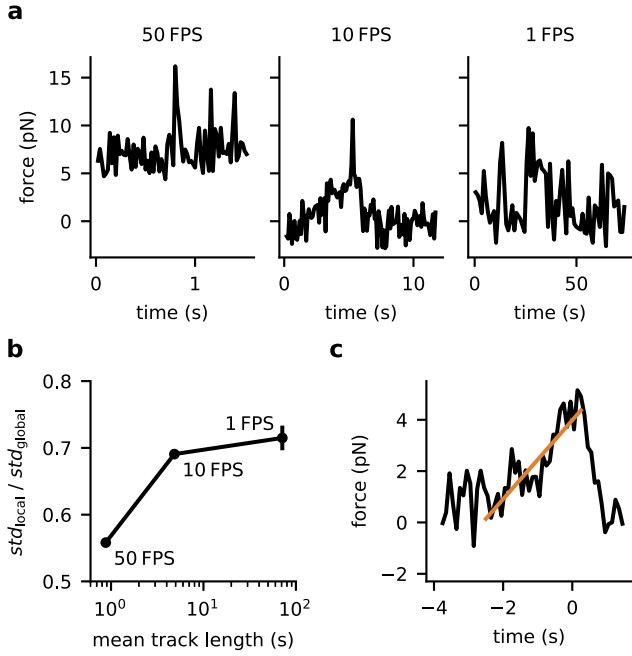

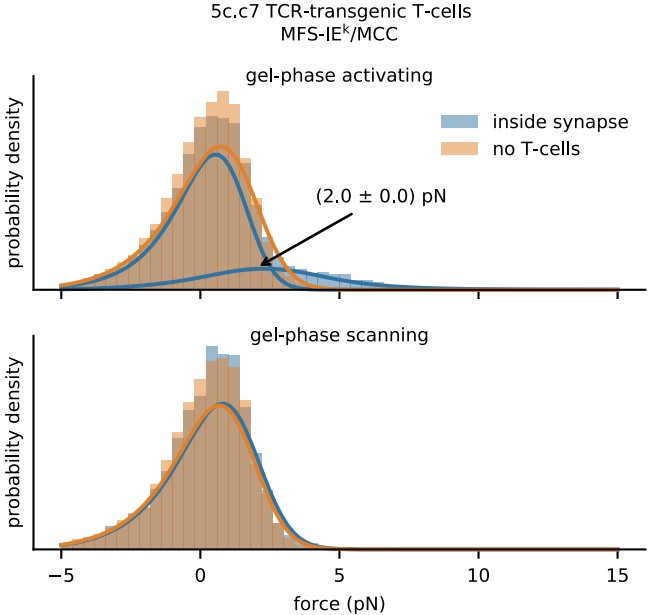

5c.c7 TCR-transgenic T-cells
MFS–IE$^k$/MCC

gel-phase activating

- inside synapse
- no T-cells

(2.0 ± 0.0) pN

gel-phase scanning

**Fig. 4 Temporal analysis of TCR-applied forces observed for activated T-cells confronted with gel-phase SLBs. a** Representative single-molecule force traces recorded at indicated frame rates. We observed transitions within force trajectories only in traces recorded at low frame rates, whereas trajectories recorded at high frame rates showed rather constant force levels. **b** For quantitative readout, we determined the force fluctuations of each trajectory as standard deviation, calculated the average over all trajectories (termed "std$_{local}$") and divided the number by the global standard deviation of all measured force values (termed "std$_{global}$") . Ergodic behavior was increasingly observed with decreasing frame rates. Error bars are standard errors of the mean. 4073 events in 94 tracks from 593 movies in dataset (50 fps), 1823 events in 38 tracks from 368 movies in dataset (10 fps), 649 events in 10 tracks from 108 movies in dataset (1 fps). **c** Assessment of the loading rate for single-molecular force transitions recorded at 10 fps. The panel displays a pooled analysis of 470 data points taken from seven clearly identifiable single-molecule force ramps, which were synchronized with respect to the force peak. A linear fit (orange line) yielded an average loading rate of 1.5 pN s$^{-1}$.

**Fig. 5 Single-molecule pulling forces exerted via single TCRs on the pMHC-functionalized molecular force sensor.** Single-molecule analysis of MFS–IE$^k$/MCC on gel-phase SLBs recorded under activating (top) or scanning conditions (bottom), using antigen-experienced 5c.c7 TCR-transgenic T-cells. We observed a significant high force peak at 2.0 ± 0.0 pN for the activating case amounting to 24% of all events, which was no longer detectable under scanning conditions. $n = 18,763$ (770 movies) for activated T-cells, $n = 14,991$ (205 movies) for stimulatory SLBs without T-cells, $n = 8646$ (348 movies) for scanning T-cells, $n = 5120$ (106 movies) for non-stimulatory SLBs without T-cells.

be ascribed to rupture of the connection between the TCR and the cytoskeleton or between the TCR and the MFS-H57.

**TCR-imposed forces on the natural ligand pMHC.** We next quantified tensile forces exerted by the TCR on stimulatory pMHC. For this we employed gel-phase SLBs decorated with the second force sensor described above for which the flagelliform peptide was connected to IE$^k$/MCC (MFS–IE$^k$/MCC) instead of H57. Functional integrity of the construct was verified by measuring the calcium response of 5c.c7 TCR-transgenic T-cell blasts (Supplementary Fig. 3C). Activating conditions gave rise to a clearly discernible peak corresponding to mean forces of 2.0 ± 0.0 pN (Fig. 5) resulting from TCR–pMHC bonds sustaining shear and pull, at least to some degree: when compared to H57–TCR binding, the majority of IE$^k$/MCC–TCR binding events showed lower tensile forces close to the MFS-based detection limit of 0.9 pN. Such differences became even more markedly evident under scanning conditions, where we no longer observed a visible high-force shoulder. Reduced forces may indicate early TCR–pMHC bond rupture prior to establishing maximal force. Comparatively low binding efficiency between MFS–IE$^k$/MCC and the TCR under scanning conditions may

further contribute to this effect, as few pulling events would be masked under the majority of non-ligated sensors. Further studies will be required to discriminate between these possible scenarios.

**Discussion**
In recent years, forces exerted on TCR–pMHC bonds have been frequently associated with T-cell antigen recognition[4–6]. The origin of these forces has been linked to actomyosin-based contractions in the case of migrating T-cells, and actin reorganization during synapse formation as a consequence of T-cell activation[41]. Indeed, in resting cells ~30% of the TCR are linked to the cytoskeleton[42]. A clutch-like coupling between the cortical actin and microcluster-resident TCRs has been proposed for their directional transport to the synaptic center following T-cell activation[43]. It is therefore conceivable that T-cells exert tensile forces onto TCR–pMHC bonds.

Our MFS-based findings are best explained by TCR-exerted forces which act largely tangentially with respect to the T-cell surface. In our T-cell stimulation regimen, fluid-phase SLBs provide an environment with substantially lower viscous drag to its embedded proteins than gel-phase SLBs. Of note, the drag $\gamma$ experienced by a lipid-anchored protein in a fluid-phase membrane is smaller than $3 \times 10^{-9}$ Pa s m[44]. As a consequence, MFSs pulled with the velocity of a TCR microcluster migrating to the synaptic center ($v = 0.14$ μm s$^{-1}$ [13]) would be subjected to a viscous force $F$ that is lower than 0.4 fN (with $F = \gamma v$) and therefore negligible when compared to $F_{min}$. Thus, moving the MFS laterally within fluid-phase SLBs does not result in a drag that is sufficient to stretch the sensor. In contrast, the viscous drag encountered in a gel-phase membrane is large enough to arrest

embedded MFSs. As a consequence, the MFS becomes invariably extended whenever the TCR places it under lateral load.

Pulling the MFS away from the SLB will result in its elongation. This is because the detectable maximum force $F_{max}$ is substantially lower than the force required to pull six to nine lipids associated with the mSAv-3xHis$_6$[45] out of the membrane (>7 pN at loading rates >2 pN s$^{-1}$ per single lipid[46] amounting to >42 pN for six parallel anchors). Furthermore, extracting a biotin from its binding pocket requires more than 25 pN at loading rates larger than 1 pN s$^{-1}$ [47]. Indeed, a recent study revealed that gel-phase lipid bilayers can sustain forces of up to 70 pN[48].

Our observations are consistent with the reported occurrence of tangential forces detected via traction force microscopy[20,21]. In this study, TCR-exerted forces of 50–100 pN were detected per pillar with a cross-sectional area of ~0.75 μm². Considering a parallel configuration of the bonds and TCR densities of ~70 molecules per μm², as they are typical for most T-cells[49], traction forces per bond would amount to a low piconewton regime, similar to the results of our study. In contrast, Ma and colleagues reported TCR-imposed forces of 4.7 pN when T-cells contacted fluid-phase SLBs[24]. However, in that study multiple DNA tension probes, and, as a consequence, multiple TCR molecules were cross-linked via the solid gold nanoparticles, allowing for tensile forces between the cross-linked TCRs.

Of particular significance for antigen recognition, we did not find any evidence for a scenario in which molecular forces beyond 2 pN exerted via the TCR are a consequence of or a precondition for T-cell signaling:

(i) We detected a high-force peak exclusively when T-cells had been seeded onto gel-phase but not fluid-phase SLBs, regardless of whether T-cells had been stimulated in the course of contact formation or not. For scanning T-cells, we attribute such forces to the contractility of the actomyosin cytoskeleton, to which plasma membrane-resident TCRs are coupled.

(ii) Furthermore, our results render it unlikely that forces larger than 2 pN and imposed on the TCR are required to trigger the TCR when T-cells are facing moderate to high densities of stimulatory ligands. After all, T-cells seeded onto functionalized fluid-phase SLBs did not give rise to changes in FRET of interacting MFS corresponding to forces higher than 2 pN even though they showed all the hallmarks of TCR-proximal signaling, such as elevated intracellular calcium levels, recruitment of ZAP70 kinase, TCR-microcluster formation, and TCR-movement towards the synaptic center. It should be noted that at this stage we cannot rule out the presence of a minority of molecules that may have experienced higher tensile forces at the onset of T-cell triggering and which would have been masked by an excess of other binding events. However, even in this hypothetical case contributions from tangential forces can be ruled out. We find it in this context noteworthy that we failed to observe any TCR-imposed forces when applying physiological ligands in the single-molecule regime (Fig. 5, second panel).

We do not state to have quantitatively accounted for initial TCR-ligand binding events, which may be critical for T-cells to commit to stable synapse formation. However, continual TCR-engagement is required for maintaining TCR-proximal signaling and the integrity of the immunological synapse as well as for unfolding the T-cell's full effector potential, even hours after cell–cell contact formation[50]. Hence, we consider the magnitude of TCR-imposed forces that we have measured in the context of antigen recognition meaningful for assessing mechanisms underlying TCR-mediated intracellular signaling.

While MFS-H57 was specifically designed in our study for its capacity to maintain TCR-bonds even under TCR-mediated drag, nominal pMHCs are typically characterized by fast TCR-dissociation requiring a significantly lower activation barrier for bond rupture: In case of IE$^k$/MCC binding to its cognate 5c.c7 TCR, already forces in the lower pN-regime appear likely to induce the dissociation of pMHC from the TCR-binding pocket, thereby reducing the average bond lifetime[4]. If TCR–pMHC dissociation is promoted by acting forces, force loads may no longer materialize to the same magnitude, as they are exerted in the case of stable TCR–antibody interactions. Our observation of reduced pulling forces in case of MFS-IE$^k$/MCC would hence be in agreement with the presence of slip bonds formed between the 5c.c7 TCR and IE$^k$/MCC, which rupture before the maximum force recordable with the use of the MFS-H57 construct has been reached.

Interestingly, our time-resolved experiments did not reveal forces that were constant but that increased linearly over time (Fig. 4c). Such force ramps should not only affect the mean but in particular the distribution of bond lifetimes, a property which could be exploited by T-cells to improve the specificity and sensitivity of antigen discrimination[4,51]. Previously, we have reasoned by analytical deduction that loading rates of only 1 pN/s would already suffice to enable peptide discrimination with high specificity and sensitivity[4].

T-cells scan the surface of the opposing APC typically via microvilli-like protrusions[30], a process that has also been described for T-cells establishing a contact with activating surfaces[30,52,53]. During this phase, the limited number of TCRs that can bind pMHC on the opposing APC become subjected to tangential and perpendicular forces upon pMHC binding and until TCR-ligand bond rupture. For tangential forces to act on pMHC-engaged TCRs within the cellular interface, MHC molecules need to be subjected to a counterforce at the level of the APC membrane, as could be inferred from the sizable number of MHC II molecules diffusing slowly within the plasma membrane of dendritic cells[54]. Since perpendicular forces measured here on fluid membranes via MFS-H57 are similar in magnitude to the average forces experienced by the MFS–IE$^k$/MCC, we consider it likely that even rather weak pulling forces exerted in a perpendicular fashion would suffice to terminate TCR–pMHC bonds. If T-cells fail to become activated while probing the APC surface, T-cells reduce pulling forces as is indicated by our temporal analysis (Fig. 3). A reduction in pulling forces over time may help to adjust towards more sensitive antigen scanning, possibly via reorganization of the actin–myosin network during the cell adhesion process. Tonic TCR-mediated signaling, which is known to provide survival signals and instruct T-cell-fate decisions in response to barely detectable low-affinity TCR–pMHC interactions, may very well depend on lowering the force regime.

A variety of mechanistic and phenomenological models[55,56] have been proposed to explain how extracellular TCR–pMHC binding is translated with high sensitivity into a ligand-specific intracellular signaling event. With rising intracellular calcium levels, LFA-1 undergoes an affinity switch to consolidate via ICAM-1-binding T-cell adhesion with the rapid formation of TCR-microclusters in the synaptic periphery. Within a microcluster multiple TCR–pMHC bonds are placed under load in parallel. Our results may hint at different consequences arising from forces applied in perpendicular and tangential pulling directions. Under activating conditions forces applied in perpendicular direction were not detectable with the use of a fluid-phase SLB, possibly reflecting force dissipation over the parallel bonds. When T-cells were confronted with gel-phase SLBs, forces imposed on individual TCRs appeared however similar under both activating and scanning conditions, implying a different

mechanism of force exertion in tangential direction. Microclusters are concomitantly transported tangentially towards the synaptic center with a broad range of velocities amounting to $20 \pm 8$ nm s$^{-1}$, a process that was reported to be strongly affected by constraints built into SLBs[43,57]: TCR-microclusters encountering barriers in an orientation perpendicular to their movement were found arrested, while barriers positioned in intermediate angles decelerated transport[43]. The latter finding can be rationalized by the loose clutch mechanism linking TCRs with the cytoskeleton[43], thereby spreading the average load over time in a make-and-break cycle. We consider it plausible that our single-molecule force traces shown in Fig. 4 represent such transient linkages, potentially formed via single motor protein molecules, and that the properties of our gel-phase SLBs prevented the transport of ligated TCR along the T-cell surface in analogy to the previously described diffusion barriers. Indeed, after force had accumulated for ~2.5 s, the connection between the TCR and cytoskeletal drag appeared to be released, resulting in an immediate drop in the force applied to the MFS. The sensor's stretching rate of 4 nm s$^{-1}$ as calculated from the observed loading rate of 1.5 pN s$^{-1}$ (for more details see the "Methods" section, force sensor calibration, Eq. (3)) corresponds to the lower end of the reported spectrum for microcluster transport rates, which may indicate reduced cytoskeletal dynamics when placed under load. In this context, the duration of microcluster formation would explain the delayed onset of pulling forces observed for T-cells encountering activating gel-phase SLBs (Fig. 3a). Further experiments will be needed to address the mechanical origin of the linear load increase. In agreement with previous findings[43] our findings rule out a direct coupling to retrograde actin flow, since the observed stretching rates of our sensors of a few nanometer per second do not comply with actin flow rates of several tens of nanometers per second[58].

In summary, we found that TCR-mediated forces occur within the immunological synapse in the 4–9 pN regime only when TCR-ligands were embedded in a gel-phase SLB, indicating their predominant shear force nature. Forces exerted on the natural ligand pMHC were of substantially smaller magnitude, and may indicate force-accelerated rupture of the TCR–pMHC interaction. In view of the afforded spatiotemporal resolution, we consider our experimental system presented here both adequate and sufficiently versatile to delineate the role and characteristics of TCR-imposed forces in TCR-proximal signaling and ligand discrimination.

## Methods

**Force sensor—anchor unit.** Recombinant mSAv-3xHis$_6$ served as anchor unit for the force sensor and was produced as described[33]. Briefly, biotin binding deficient subunits equipped with a C-terminal His$_6$-tag—"dead"—and functional subunits with C-terminal 3C protease cleaving site followed by Glu$_6$-tag—"alive"—were expressed as inclusion bodies in *E. coli* BL-21 using the pET expression system (Novagen). Inclusion bodies were dissolved in 6 M guanidine hydrochloride at 10 mg/ml, mixed at a ratio of 3 "dead" to 1 "alive" subunit and refolded in 100 volumes PBS. The salt concentration was reduced to 2 mM by cycles of concentration using a 10 kDa cutoff spin concentrator (Amicon) and dilution with 20 mM Tris pH 7.0. The mixture was loaded onto a MonoQ 5/50 column (GE Healthcare), and after washing the column with 100 mM NaCl in 20 mM Tris pH 7.0 mSAv-3xHis$_6$ was eluted at 20 mM Tris pH 7.0, 240 mM NaCl. Human 3 C protease (Pierce) was added and allowed to cleave the Glu$_6$-tag overnight. The digested protein was subjected to Superdex 200 (GE Healthcare) size exclusion chromatography in PBS to yield to pure anchor unit.

**Force sensor—spring unit.** The MFS peptide biotin-N-GGCGS(GPGGA)$_5$GG-KYGGS-K(ε-N$_3$) (residues for fluorophore attachment are underlined), MFS$_C$ peptide biotin-N-GGGGS(GPGGA)$_5$GGKYCGS-K(ε-N$_3$) and the MFS$_0$ peptide biotin-N-GGGGS (GPGGA)$_5$ GGKYG GS-K(ε-N$_3$) were purified via preparative C18 reverse phase HPLC (5 μm particle size, 250 × 21.2 mm, Agilent) applying a linear gradient of 0.1% trifluoracetic acid (TFA) in deionized water (buffer A) and 0.1% TFA, 9.9% deionized water, 90% acetonitrile (buffer B) over 80 min at a flow rate of 5 ml/min. The identity of the spring modules was confirmed via

MALDI-TOF mass spectrometry (Bruker). 350 μg lyophilized peptide was dissolved in PBS at a final concentration of 5 mg/ml, mixed with Alexa Fluor 555 maleimide (Thermo Fisher Scientific) at a 1.5-fold molar surplus and then incubated for 2 h followed by an incubation with 1.5-fold excess Alexa Fluor 647 succinimidyl ester (Thermo Fisher Scientific) in the presence of 330 mM NaHCO$_3$ for 2 h at room temperature. HPLC purification and lyophilization was performed after each step and successful conjugation was confirmed via mass spectrometry.

**Force sensor—functional unit.** TCRβ-reactive H57 antibody single chain fragments featuring a C-terminal unpaired cysteine residue (H57 cys scF$_V$) were expressed in *E. coli* BL-21 as inclusion bodies using the pET expression system (Novagen) and refolded by multiple dialysis steps continuously diluting the concentration of guanidine hydrochloride within the refolding buffer (50 mM Tris pH = 8.0, 200 mM NaCl, 1 mM EDTA); in a final step the refolded protein was dialyzed against 1×PBS[9,42]. The product was then concentrated using a 10 kDa cutoff spin concentrator (Amicon) in the presence of 25 μM Tris-(2-carboxyethyl)phosphine hydrochloride (TCEP, Pierce). Monomeric H57 cys scF$_V$ was purified by Superdex 75 (GE Healthcare) size exclusion chromatography in PBS, then concentrated to 1 mg/ml in the presence of 25 μM TCEP and allowed to react with a 10-fold surplus of dibenzylcyclooctyne-maleimide (DBCO, Jena Bioscience) for 2 h at room temperature. Unreacted DBCO was removed by gel filtration and proteins were brought to a concentration of 1 mg/ml.

IE$^k$ α featuring an additional C-terminal cysteine residue and IE$^k$ β were also expressed as inclusion bodies and refolded in the presence of MCC 88–103 peptide (ANERADLIAYLKQATK, Intavis) for 2 weeks[9]. Correctly folded IE$^k$ molecules were isolated via an in house built 14-4-4 antibody column, reduced with 25 μM TCEP followed by S200 gel filtration. Monomeric IE$^k$ MCC with unpaired cysteine residue was collected and coupled to DBCO as described above.

**Force sensor assembly.** Refolded from *E. coli* inclusion bodies H57-scF$_V$ were modified for sensor attachment with DBCO maleimide at a mutant unpaired C-terminal cysteine residue (Supplementary Fig. 1A–D). MFS peptides were mixed with DBCO-conjugated proteins and 25 vol% 2 M Tris pH 8.0 at a molar ratio of at least 5:1 and incubated for 2 h at room temperature. Unreacted protein was removed with the use of a monomeric avidin column (Pierce). After elution with biotin, unconjugated spring was separated from MFS conjugates using size exclusion chromatography on a Superdex 75 column (GE Healthcare). mSAv-3xHis$_6$ gel shift assays documented successful conjugation as only a neglectable proportion of H57-scFv gave rise to dimers (Supplementary Fig. 1E). The existence of such entities may have yielded a small underestimation in ensemble readouts. However, in the case of single-molecule experiments, multimeric structures could be directly identified and subsequently eliminated from the analysis.

**T-cells.** Antigen-experienced 5c.c7 murine T-cell blasts were obtained as previously described[50]. Briefly, 7.5*10$^6$/ml splenocytes were stimulated with 2 μM HPLC-purified MCC peptide (ANERADLIAYLKQATK, Intavis Bioanalytical Instruments) in T-cell medium, which consisted of RPMI 1640 (Gibco) supplemented with 10% FCS (Sigma), 100 U/ml penicillin/streptomycin (Gibco), 2 mM glutamate (Gibco), 1 mM sodium pyruvate (Gibco), 1× non-essential amino acids (Gibco) and 50 μM ß-Mercaptoethanol (Gibco). Culture volume was doubled and also supplemented with 100 U/ml IL-2 (eBioscience) on day 2. T-cell cultures were expanded in a ratio of 1:1 on day 3 and 5. Dead cells were removed by centrifugation on a cushion of Histopaque-1119 (Sigma) on day 6. T-cells were used for experiments on day 7–9 after initial stimulation.

OT-1 αβ TCR-transgenic mice of 8–12 weeks were sacrificed to collect spleen and lymph nodes (housing conditions: 22 °C, 53% relative humidity, 12 h dark–light cycle). Isolated splenic APCs were loaded with 1 μM C18 reverse-phase HPLC-purified Ovalbumin (Ova 257–264, SIINFEKL) peptide. After 1 h incubation, splenic APCs were washed with RPMI 1640 culture medium (Gibco). Subsequently, lymph-node-derived cells were mixed in a 1:2 ratio with OVA-peptide pulsed splenocytes and cultured in RPMI 1640 culture medium (Gibco) supplemented with 10% fetal calf serum (FCS) (Sigma), 100 U/ml penicillin/ streptomycin (Gibco), 2 mM glutamate (Gibco), 1 mM sodium pyruvate (Gibco), 1× non-essential amino acids (Gibco), and 50 μM ß-mercaptoethanol (Gibco) at 37 °C and 5% CO$_2$. On day 2, IL-2 was added to a final concentration of 50 U/ml. Cells were split 1:1 with fresh media every second day (day 3, 5, and 7). Dead cells were removed on day 7 prior to splitting by Ficoll-Paque Plus (Merck) step gradient centrifugation and cytotoxic T lymphocytes (CTLs) were used on day 8–10 for experiments.

**Ethical compliance statement.** Mice were bred and euthanized for T-cell isolation according to guidelines and protocols evaluated by the ethics committee of the Medical University of Vienna and approved by the Federal Ministry of Science, Research and Economy, BMWFW (BMWFW-66.009/0378-WF/V/3b/2016). Animal husbandry and experimentation were performed under national laws (Federal Ministry of Science, Research and Economy, Vienna, Austria) and the ethics committee of the Medical University of Vienna and according to the guidelines of the Federation of Laboratory Animal Science Associations (FELASA).

**Reagents**. Hank's buffered salt solution (HBSS) and phosphate buffered saline (PBS), Cytochalasin D (from Zygosporium masonii) as well as fetal calf serum (FCS) were from Sigma Aldrich (Merck KGaA, Germany). DPPC, 1-palmitoyl-2-oleoyl-glycero-3-phosphocholine (POPC), and 1,2-dioleoyl-sn-glycero-3-[(N-(5-amino-1-carboxypentyl)iminodiacetic acid)succinyl] (nickel salt) (Ni-NTA-DGS) were from Avanti Polar Lipids, Inc., USA. ICAM-1 and B7-1 were produced in Hi-5 insect cells using baculovirus and purified via size exclusion and ion exchange chromatography[9].

**Preparation of functionalized lipid bilayers**. 98 mol-% POPC or DPPC and 2 mol-% Ni-NTA-DGS, all dissolved in chloroform, were mixed and then dried under nitrogen flow for 20 min. The lipid mixture was suspended in 1 ml PBS at room temperature (POPC) or 55 °C (DPPC) and then sonicated for 10 min in an ultrasound bath (USC500TH, VWR, England) at the same temperature to form small unilamellar vesicles (SUVs). The SUV solution was finally diluted to 125 μM using PBS.

Cover slips (MENZEL-Gläser Deckgläser 24 × 60 mm #1.5) were treated in a plasma cleaner (PDC-002 Plasma Cleaner Harrick Plasma, Ithaca, NY, USA) for 10 min and glued to the bottom of an eight-well chamber (Nunc Lab-Tek, Thermo Scientific, USA) using duplicating silicone (Twinsil soft 18, picodent, Germany). SUVs were incubated for 20 min at room temperature (POPC) or 55 °C (DPPC) to form bilayers. After incubation, excess vesicles were washed away using PBS.

His-tagged proteins (for activating conditions: 10 ng mSAv-3xHis$_6$, 30 ng ICAM-1, 50 ng B7-1; for scanning conditions: 0.4 ng mSAv-3xHis$_6$, 30 ng ICAM-1) were added in 0.5 ml PBS to each well and incubated for 75 min at room temperature. Afterwards, the bilayer was washed with PBS.

Finally, the bilayer was incubated for 20 min with biotinylated MFS variants (for activating conditions: 10 ng MFS$_0$, 5–20 pg MFS or MFS$_C$; for scanning conditions: 20–60 pg MFS or MFS$_C$) in 0.5 ml PBS per well for binding to the mSAv-3xHis$_6$. Bilayers were washed again to remove excess MFS.

Immediately before adding cells, the buffer was exchanged for HBSS containing 2% FCS.

For ensemble experiments, molecular densities were determined by dividing the fluorescence signal per pixel by the single molecule brightness recorded at the same settings, considering the effective pixel width of 160 nm. In single molecule samples, densities were determined by counting the number of molecules and dividing by the area of the field of view.

**T-cell calcium imaging**. 3 ml of HBSS + 2% FCS were added to ~$10^6$ T-cells. After centrifugation (300×g for 3 min), supernatant was removed and cells were incubated with 1 μg of Fura-2 AM dye (Thermo Scientific, USA) at 1 mg/ml DMSO in a total volume of 100 μl for 20 min at room temperature, and washed (addition of 5 ml HBSS + 2% FCS, centrifugation at 300×g for 3 min, removal of supernatant, resuspension in 100 μl). Cells were kept at room temperature until seeding them onto the functionalized bilayers and for a maximal duration of 30 min.

Time-lapse movies were recorded using alternating excitation (340 and 380 nm, 50 and 10 ms illumination, respectively) using a monochromatic light source (Polychrome V, TILL Photonics), coupled into a Zeiss Axiovert 200M microscope equipped with a 10× objective (UPlanFL N 10x/0.30, Olympus) and an Andor iXon Ultra 897 EMCCD (Andor Technology Ltd, Belfast, UK) camera. The total recording time was 15 min or longer at 1 Hz.

Cells were tracked in each frame using a particle tracking algorithm published by Gao and Kilfoil[59]. Tracking parameters were chosen accordingly to only include cells in contact with the SLBs. A custom-built MATLAB software was used to generate the ratio images for each frame. We used the average ratio of each cell in a central circle of 2 μm.

Population analysis was performed using a custom-build MATLAB software based on the "Methods for automated and accurate analysis of cell signals"[60]. Briefly, all ratio traces of a population were normalized frame-wise to the population median of the negative control. Cells with at least 80% of their trajectory above activation threshold were counted as activated. Individual trajectories were aligned to the maximum ratio value of each trajectory creating a synchronized population with respect to peak position. The population was synchronized with respect to cell adhesion and contained individual cellular trajectories starting from cell contact with the SLB. The data was displayed as medians with interquartile range. As positive control we used SLBs decorated with 30 ng ICAM-1, 50 ng B7-1 and 10 ng His-tagged MCC-IE$^k$ amounting to ~100 molecules μm$^{-2}$, as negative control SLBs decorated with only ICAM-1.

**T-cell fixation and immunostaining**. Prior to T-cell seeding, ligand densities on SLBs were determined as described above. 5 c.c7 TCR-transgenic T-cells were seeded onto SLBs and allowed to settle for 10 min at room temperature. Fixation buffer (PBS, 8% formaldehyde (28908, ThermoFisher Scientific), 0.2% glutaraldehyde (G7776-10ML, Sigma Aldrich), 100 mM Na$_3$O$_4$V (S6508-10G, Sigma Aldrich), 1 M NaF (S7920-100G, Sigma Aldrich)) was added 1:1 and incubated for 20 min at room temperature, before samples were rinsed with PBS. Permeabilization buffer (PBS, 100 mM Na$_3$O$_4$V, 1 M NaF, 0.2% Triton-X (85111, Thermo-Fisher Scientific)) was added and after 1 min samples were washed with PBS before blocking with passivation buffer (PBS, 3% BSA) for 30 min. Anti-ZAP70 antibody

labeled with AF647 (clone 1E7.2, #51-6695-82, ThermoFisher Scientific) was added at a final concentration of 1.25 μg/ml and incubated overnight at 4 °C. Samples were washed extensively with passivation buffer. Imaging was performed in HBSS.

**Inhibition of actin polymerization**. 5c.c7 T-cells were seeded onto SLBs in HBSS containing 2% FCS and incubated for 7 min at room temperature. 10 μM Cytochalasin D in DMSO was carefully added and was incubated for 10 min at room temperature prior to single-molecule FRET measurements. A mock sample was incubated with the same volume of DMSO. Effect of inhibition was monitored in parallel by calcium flux measurements.

**Microscopy experiments**. Experiments were performed at room temperature and carried out on two different microscopy systems.

In the first microscopy system, fluorophores were excited with 640 nm (OBIS 640, Coherent, CA) or 532 nm (Millenia Prime, Spectra Physics, CA) laser light coupled into an epifluorescence microscopy (Zeiss, Germany). All experiments were performed in objective-type TIR configuration, using a high-NA objective (α Plan-FLUAR 100x/1.45 oil, Zeiss, Germany) and a quad-band dichroic mirror (Di01-R405/488/532/635-25×36, Semrock Inc., USA) to separate excitation light from emission light. The fluorescence emission path was split along wavelengths with the use of an emission beam splitter system (Optosplit II, Cairn Research, Kent, UK) employing a dichroic mirror with a cutoff wavelength of 640 nm (FF640-FDi01-25 × 36, Semrock Inc., USA) and emission filters (ET570/60m and ET675/50m band pass, Chroma Technology Corp, USA). Both color channels were imaged side-by-side on the same chip of an EM-CCD camera (Andor iXon Ultra 897, Andor Technology Ltd, UK). In addition, we used a 405 nm laser (iBeam Smart 405-S, Toptica Photonics AG, Germany) for visualizing the cell contours.

In the second microscopy system, excitation was achieved by coupling a 532 nm (OBIS LS, Coherent, USA) and a 640 nm (iBeam smart, Toptica Photonics, Germany) laser line into a Ti-E inverted microscope (Nikon, Japan) via a dichroic mirror ZT405/488/532/640rpc (Chroma, USA) into a ×100 objective (SR Apo TIRF, Nikon, Japan) in an objective-based TIR setting. Emission was split using an Optosplit II (Cairn Research, UK) equipped with a 640 nm dichroic mirror (ZT640rdc, Chroma, USA) and emission filters (ET575/50, ET655LP, Chroma, USA) and simultaneously imaged on an Andor iXon Ultra 897 EM-CCD camera (Andor Technology, UK). Calcium imaging was performed using a Lambda LS xenon lamp (Sutter Instruments, USA) with excitation filters (340/26, 387/11, Semrock, USA) for illumination via a ×20 objective (S FLUOR, Nikon, Tokyo, Japan) and Fura-2 emission filter (ET525/50, Chroma, USA). Devices were controlled by MetaMorph imaging software (Molecular Devices, USA).

Microscopy system #1 was used for all single molecule experiments and for ensemble FRET measurements with the use of DPPC bilayers. Microscopy system #2 was employed for ensemble FRET experiments using POPC membranes and for calcium measurements.

**Ensemble FRET measurements**. FRET efficiencies were determined at the ensemble level using DRAAP. For this, an image of the donor channel was recorded before and after acceptor photobleaching, and the pixel-averaged fluorescence signal $f_{pre}$ and $f_{post}$, respectively, was calculated for each cell. The FRET efficiency was then given by $E_{bulk} = (f_{post} - f_{pre})/f_{post}$. For quantitation an area was selected underneath the T-cell and averaged, giving rise to $E_{bulk}^{cell}$. The same area was also recorded on lipid bilayers devoid of T-cells, analyzed for FRET efficiency, and averaged, yielding $E_{bulk}^{no\ cell}$. Finally, the difference was calculated via $\triangle E_{bulk} = E_{bulk}^{cell} - E_{bulk}^{no\ cell}$. Bleaching times were between 500 ms and 1 s, illumination times and delays were typically below 10 ms. Single-molecule FRET measurements

**Single-molecule FRET measurements**. In a typical experiment we recorded 300 images alternating between green and red excitation, with illumination times $t_{ill}$ = 5 ms (see Supplementary Fig. 5B). The delay between the green and the red color channel, $t_{gr}$, was kept as short as possible, typically $t_{gr}$ = 5–15 ms. The delay between two consecutive green-red doublets, $t_{delay}$, was chosen in a range between 5 ms and 975 ms. For each time point, three images were recorded: donor emission upon donor excitation (Im$_{DD}$), acceptor emission upon donor excitation (Im$_{DA}$), and acceptor emission upon acceptor excitation (Im$_{AA}$). To convert camera counts into emitted photons, count values were multiplied by the manufacturer-provided camera efficiency at the chosen settings (15.7 photons per count) and divided by the EM gain (typically, 300).

To determine single-molecule FRET efficiencies, a series of seven processing steps was performed (Supplementary Figs. 5 and 10). Data analysis was implemented in Python 3, using the Numpy, SciPy[61], and Pandas packages for general numerical computations, matplotlib for plotting and data handling[62], scikit-learn for fitting of Gaussian mixture models[63], trackpy for single-molecule tracking[64], pims for reading image sequences from disk, and OpenCV for adaptive thresholding. Jupyter notebook and ipywidgets were used as a front-end to Python.

As a first step, two-color images (Im$_{DD}$ and Im$_{DA}$) were registered and corrected for chromatic aberrations using beads of 200 nm diameter (TetraSpec, Invitrogen) as fiducial markers, by fitting an affine transformation to the set of

localization pairs. In our case, we transformed and interpolated the donor color channel to match the acceptor color channel ($T(\mathrm{Im_{DD}})$).

Secondly, we identified candidates for single-molecule signals. To this end, we localized single-molecule signals on the sum image $T(\mathrm{Im_{DD}}) + \mathrm{Im_{DA}}$ and on the image $\mathrm{Im_{AA}}$[65], and interlaced the resulting localizations according to the recording sequence. Taking the sum image ensured the detection of both low FRET and high FRET signals with high sensitivity. The resulting single-molecule position data were linked into trajectories using a single-molecule tracking algorithm[64], with a maximum search radius of 800 nm between two consecutive frames. Only trajectories with a length ≥ 4 observations were considered. We allowed for gaps of up to three frames in the trajectories to account for bleaching and blinking of fluorophores as well as missed signals. These gaps were filled using linear interpolation between the corresponding positions before and after the missed events. As a result, we obtained the coordinates and according trajectories of all candidates for single-molecule signals in the coordinate system of the acceptor channel. For further processing, the coordinates corresponding to signals recorded in the donor channel were transformed back to the coordinate system of the donor channel via $T^{-1}$.

Next, we determined the fluorescence brightness values. We first applied a Gaussian blur ($\sigma = 1$ pixel) to the three raw data images $\mathrm{Im_{DD}}$, $\mathrm{Im_{DA}}$, and $\mathrm{Im_{AA}}$, which reduces noise while preserving the signal intensity (Supplementary Fig. 5D). To determine the single-molecule brightness, the camera counts in a 9px-diameter foreground circle around each signal's position, $s_{ij}$, were considered (Supplementary Fig. 5C). Local background, bg, was estimated by calculating the mean camera count in a ring around the signal circle (excluding any pixels affected by nearby signals). Fluorescence brightness was calculated according to $f = \sum_{i,j\in\mathrm{circle}}(s_{ij} - \mathrm{bg})$. Finally, f was corrected for uneven excitation laser profile across the field of view $f \longmapsto f(x,y)/\mathrm{profile}(x,y)$. We determined $\mathrm{profile}(x,y)$ for each laser from heavily stained samples, applied a Gaussian blur ($\sigma = 10$px), and scaled to a maximum value of 1. For each signal, we hence obtained here three values: the fluorescence brightness at donor excitation in donor emission ($f_{\mathrm{DD}}$), donor excitation in acceptor emission ($f_{\mathrm{DA}}$), and acceptor excitation in acceptor emission ($f_{\mathrm{AA}}$).

We applied several initial filter steps to disregard signals which do not match stringent criteria for single-molecules. (i) remove all signals from areas with <50% of the maximum laser intensity in the donor channel. (ii) remove all trajectories that show potential overlap with other trajectories. Trajectories, where >25% of data points showed overlapping foreground circles, were completely disregarded. In all other trajectories, only single signals with overlapping foreground circles were removed. (iii) remove all trajectories which are not present in the first frame. This step ensures that the full photobleaching profile of each trajectory is available in the subsequent steps. (iv) single step transitions were identified independently in the time trace of $f_{\mathrm{AA}}$ and of $f_{\mathrm{DD}} + f_{\mathrm{DA}}$[66]. Only trajectories were selected for further processing where $f_{\mathrm{AA}}$ showed single step photobleaching. Also, trajectories showing partial photobleaching in $f_{\mathrm{DD}} + f_{\mathrm{DA}}$ were discarded.

As a next step, we calculated the corrected single-molecule FRET efficiency $E$ and the FRET stoichiometry factor $S$ according to refs. [40,67]:

$$E = \frac{f_{\mathrm{DA}} - \alpha f_{\mathrm{DD}} - \delta f_{\mathrm{AA}}}{\gamma f_{\mathrm{DD}} + f_{\mathrm{DA}} - \alpha f_{\mathrm{DD}} - \delta f_{\mathrm{AA}}} \quad (1)$$

$$S = \frac{\gamma f_{\mathrm{DD}} + f_{\mathrm{DA}} - \alpha f_{\mathrm{DD}} - \delta f_{\mathrm{AA}}}{\gamma f_{\mathrm{DD}} + f_{\mathrm{DA}} - \alpha f_{\mathrm{DD}} - \delta f_{\mathrm{AA}} + \frac{f_{\mathrm{AA}}}{\beta}} \quad (2)$$

The bleedthrough coefficient, $\alpha$, and the coefficient describing direct acceptor excitation, $\delta$, were determined from separate experiments using donor only and acceptor only. The detection efficiency, $\gamma$, was determined globally from single step photobleaching events according to ref. [68] using MFS$_{\mathrm{C}}$ datasets without cells as they provide large intensity differences due to high FRET efficiencies. The excitation efficiency factor $\beta$ scales $f_{\mathrm{AA}}$; it is adjusted to reach $S = 0.5$ in case of a calibration sample of known stoichiometry[69]. In our case, we used MFS without cells as calibrations standard.

After the correction, we applied further filtering steps. (i) only data were taken before the bleaching step of the acceptor (or donor, if donor bleaching happened prior to acceptor bleaching). (ii) to exclude incorrectly synthesized force sensors or clusters in the data analysis, we removed trajectories where more than 25% of the corresponding data showed $S > 0.6$ (excluding two or more donors per acceptor) or $S < 0.35$ (excluding two or more acceptors per donor). (iii) for experiments with cells, we selected only data points underneath an adhered T-cell. We identified T-cells from Fura-2 images, which were recorded synchronously to the FRET signals, using an adaptive thresholding algorithm.

Finally, 2D S-E data were fitted with a maximum-likelihood fit, assuming a Bayesian Gaussian mixture model[63]. In case of a low FRET peak exceeding the maximum detectable FRET value $E_{\mathrm{max}}$ (see subsection "Dynamic range of the force sensor") or if its relative weight was smaller than 0.1 the results of a single component Gaussian fit were shown.

For determination of the standard error, we used bootstrapping analysis. Briefly, 100 samples (identical in size to the original data) were drawn with replacement and fitted. Specified error bars correspond to the standard deviations of the fit results.

**Calibration of the force sensor**. The calibration of the force sensor is based on the study by Brenner et al. [29], who found for the same flagelliform protein a linear force–distance relationship according to

$$F = \frac{r - b \cdot n - c}{a \cdot n} \quad (3)$$

where $r$ is the separation of the two fluorophores, $b = 0.044$ nm the average length of a single amino acid in the collapsed state, $c$ a constant linker length, and $a = 0.0122$ nm pN$^{-1}$ the compliance per amino acid. $n = 25$ was the number of amino acids between the two fluorophores (in our case $n = 29$). From the $r$-dependence of the FRET efficiency,

$$E = \frac{1}{1 + \left(\frac{r}{R_0}\right)^6} \quad (4)$$

we determined the functional dependence of $F$ on $E$. Here, $R_0 = 5.1$ nm denotes the Förster distance for the FRET pair AlexaF647 and AlexaF555[70]. The parameter $c$ in Eq. (3) can be determined from experiments with $F = 0$, which were realized here by recording MFS on a supported lipid bilayer without cells; from the according FRET efficiency $E$, we calculated $c = 2.4$ nm.

Throughout the paper, we specify the expectation value of the fit results for the single molecule force distributions, which corresponds to the mean value of the observed pulling forces.

**Determination of the loading rate**. Single-molecule trajectories recorded on DPPC bilayers under activating conditions at 10fps were selected for transitions between low and high force levels, based on visual inspection (see Fig. 4a, second panel for a representative example). The selected trajectories were aligned along the time axis such that the peak forces coincided, and the rising phase was fitted with a linear function. For display, single-molecule forces were averaged for each time point.

**Dynamic range of the force sensor**. Simulations were implemented in Python 3. The NumPy package was used for general numerical operations as well as for random number generation.

To assess the dynamic range of the force sensor, we based our calculations on the reported elastic properties of the flagelliform protein[29]. As a conservative estimate of $E_{\mathrm{min}}$, we considered the MFS to follow a linear force-extension relationship up to 10 pN tensile forces. Using Eqs. (3) and (4) this relates to a minimum FRET efficiency $E_{\mathrm{min}} = 0.11$.

$E_{\mathrm{max}}$ is limited by the difficulty to discriminate the low-FRET peak from the high-FRET peak. To quantify $E_{\mathrm{max}}$, we simulated Gaussian-distributed random data sets for the high-FRET peak, taking a mean FRET efficiency of 0.87 and a standard deviation $\sigma = 0.12$, as typically observed throughout the experiments. For the low FRET peak, we simulated varying values for the mean FRET efficiency and the standard deviation at a weight of 0.15 or 0.25; the weight of 0.25 reflect the experimental data obtained with MFS-H57 on fluid-phase membranes under scanning conditions (Supplementary Fig. 7), the weight of 0.15 is a conservative estimate for a challenging scenario. Samples consisting of 2300 data points were simulated and compared to a distribution lacking the low FRET peak, using a two-sample Kolmogorov–Smirnov test. The determined $p$-values are shown in Supplementary Fig. 5A (corresponding values in Supplementary Table 1): regions of low $p$-values correspond to parameter settings which allow for identification of a low FRET peak. As expected, the smaller the weight of the low FRET becomes, the more difficult is its identification. As a conservative estimate we took for the estimate of the dynamic range a weight of 0.15, yielding a threshold of approximately $E_{\mathrm{max}} = 0.8$, which corresponds to $F_{\mathrm{min}} = 0.9$ pN.

**Simulation of single-molecule signals**. To assess the quality of the Gauss blur (Supplementary Fig. 5D), we simulated Gauss-distributed signals on a grid, with the $\sigma$-width of the Gauss being identical to the pixel size. On each pixel, the obtained number was Poisson-distributed (to simulate shot noise), and camera noise was added by a normal distributed value, which was taken from recordings on our microscopy setup. The obtained signals were analyzed as described above, both with and without the Gauss blur.

**Diffusion analysis**. To obtain diffusion coefficients in Supplementary Fig. 2A, image sequences were acquired at an illumination time of $t_{\mathrm{ill}} = 5$ ms and a delay of $t_{\mathrm{delay}} = 15$, 95, or 995 ms. Single-molecule positions were determined as for FRET analysis.

Mean square displacement analysis was performed by first calculating the squared step lengths for all localizations in all tracks from each frame to the next, from each frame to the one after the next, up to a span of 10 frames, yielding a set of squared displacements for each lag time $t_{\mathrm{lag}} =$ number of skipped frames × ($t_{\mathrm{ill}} + t_{\mathrm{delay}}$). For each set, 100 rounds of drawing randomly with replacement (bootstrapping) and calculating the mean value were performed. In every round, the equation MSD $= 4Dt_{\mathrm{lag}} + 4\varepsilon^2$ was fit to determine the diffusion coefficient $D$ and the localization uncertainty $\varepsilon$. The provided results

are the mean values ± the standard deviations of the 100 runs, i.e., the standard errors of the statistics in question[71].

**Analysis of the mobile fraction**. To determine the immobilized molecules (Fig. 2b and Supplementary Fig. 8), we calculated the smallest enclosing circle for each trajectory using the Python implementation from Project Nayuki (https://www.nayuki.io/page/smallest-enclosing-circle). Since diffusion coefficients scale with the radius of this circle, $r_c$, over the square-root of the length of the trajectory, $\sqrt{t_d}$, we plotted histograms of $r_c/\sqrt{t_d}$, yielding peaks for the immobilized molecules $r_c/\sqrt{t_d} < 0.35\,\mu\text{m} \cdot \text{s}^{-0.5}$ and mobile molecules $r_c/\sqrt{t_d} \geq 0.35\,\mu\text{m} \cdot \text{s}^{-0.5}$.

**Statistical analysis**. Statistical analysis was performed with Mann–Whitney $U$ tests to evaluate differences in the mean values of the distributions. $p$-values < 0.01 were considered as significant, ≥0.01 as not significant (n.s.). In the figure legends, we provide the number of the analyzed movies contributing to the corresponding data set, where each movie comprises between 1 and 4 T-cells.

**Reporting summary**. Further information on research design is available in the Nature Research Reporting Summary linked to this article.

## Data availability

Data supporting the findings of this manuscript are available from the corresponding authors upon reasonable request. A reporting summary for this Article is available as a Supplementary Information file. Source data are provided with this paper.

## Code availability

We provide the Python code for single-molecule FRET analysis[72] as well as the underlying Python library[73].

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

## Acknowledgements

This work was supported by the Austrian Science Fund (FWF) projects P30214-N36 (L.S., G.J.S.), P32307-B (J.G., G.J.S.), and P 25775-B2 (F.K., J.B.H.), the Boehringer Ingelheim Fonds (R.P.), Long-Term Fellowship from Federation of European Biochemical Societies (E.K.), Human Frontiers Science Program RGY0065/2017 (E.K.), and by the Vienna Science and Technology Fund (WWTF) LS13-030 (J.G., F.K., L.S., J.B.H., G.J.S.). We thank Vanessa Mühlgrabner for isolation and preparation of primary mouse T-cell blasts.

## Author contributions

J.G. and F.K. synthesized constructs. J.G., F.K., and L.S. performed imaging experiments. L.S. developed single-molecule data analysis tools. H.S. and R.P. provided important insights. G.J.S., J.B.H., and E.K. conceived the study. J.G., F.K., L.S., J.B.H., and G.J.S. contributed to data analysis, interpretation of the results and wrote the manuscript.

## Competing interests

The authors declare no competing interests.
