## [Peer Review File · Nature Communications]

Reviewer #1 (Remarks to the Author):

Summary:

This manuscript describes the development and application of a molecular tension sensing technique that allows the quantification of mechanical force across the T-cell receptor (TCR) in a single-molecule resolved fashion. The authors functionalized supported lipid bilayers (SLB) with a FRET-based tension sensor carrying a TCR-reactive single chain antibody. They seeded 5c.c7-derived T-cells onto these surfaces, and then started by performing FRET ensemble measurements. These experiments demonstrate the functionality of the approach indicating pN-scale forces across the T-cell receptor on gel-like SLB that allow for shear- and pulling forces. By contrast, forces are not observed on fluid-like SLB that are characterized by high lateral mobilities and do not allow the generation of tangential forces.

Next, the authors dilute the amount of tension sensor molecules in the SLB to perform single-molecule resolved force measurements. These experiments confirm the previous observations (i.e. tension on gel-like SLB but no or little force on fluid-like SLB) and reveal forces of about 4–5 piconewton (pN) per TCR-antibody linkage on gel-like SLB. An unexpected finding is that forces appear to be largely independent of the T-cell activation state. Finally, the authors perform a temporal analysis of the single-molecule FRET trajectories to determine an average loading rate across the TCR-linkage of about 1.5 pN/s.

Overall, this is a technically interesting study. Even though FRET-based tension sensors have already been applied to investigate molecular forces with single-molecule resolution in cells (Morimatsu et al 2013, Nano Lett), single-molecule-resolved force measurements have (to my knowledge) not been realized with molecular force sensors in T-cells before, which makes this study quite intriguing. There is no doubt that the physiological background is highly relevant, but the degree of biological insight is quite limited at this point, in part because a study using a similar force sensing approach to study T-cell mechanics in ensemble measurements has been published before (Liu et al, 2016, PNAS). More importantly, the here established approach seems to provide only limited insights into the mechanics of physiologically occurring TCR–pMHC interactions. Mechanisms of regulation by co-receptors or cytoskeletal networks are also not investigated, and important external factors that are thought to modulate T-cell mechanics (e.g. shear flow) are not considered. I also feel that the T-cell activation experiments should be evaluated/controlled more rigorously, and the authors should improve the way the data are presented. I would like to encourage the authors to address these issues and thereby increase the impact of this study. For more detailed comments, please see below.

Major points:

1. Owing to the nature of the here established setup, the measurements do not probe a naturally occurring interaction between the TCR and the pMHC, but an artificial linkage mediated by the TCR and a single-chain antibody (H57-scFv). It is therefore unclear to which degree the here measured forces reflect the physiological situation, where highly specific pMHC–TCR interactions trigger the T-cell response. This seems to inherently limit the study, especially because the central hypothesis in the field of T-cell mechanics, that agonist-bound MHC-TCR interactions form catch-bond-like linkages (Liu et al, 2014, Cell), cannot be directly tested. I am not sure whether the authors can find a way to improve the specificity of their setup, but it would significantly increase the impact of this paper if more physiologically relevant measurements were possible.
2. The authors have acknowledged that previous work has been published, which is conceptually quite similar to the here used approach (Liu et al, 2016, PNAS). I do not think that this is a problem per se, because this study here addresses T-cell mechanics on a different scale (i.e. single-molecule resolution). I also agree with the authors that the loading rate sensitivity of DNA-based tension

sensors is quite problematic when it comes to the calculation of molecular forces. Nevertheless, the authors should provide much more biological experiments to document the advantages of the here established technique. Such experiments could include the use of different T-cell subtypes, exploring effects of distinct T-cell co-receptors (e.g. CD8/CD8, ICAM, integrins), or the application of shear flow. The authors should also use inhibitors to test whether the observed forces across the TCR on gel-like substrates are subject to cytoskeletal regulation (Thauland et al. 2017, Sci Signal).

3. An intriguing finding of this study is that very small forces support T-cell stimulation. A critical data set for this interpretation is Supp. Fig. 2B showing data from ratiometric calcium imaging. These data sets demonstrate that adding molecular force sensors at sufficiently high concentrations to SLB can induce a Ca²⁺ response. However, the response is clearly different from the IEK-MCC-induced condition, which leads to a quicker (gel-like SLB) or more potent (fluid-like SLB) response. The question therefore is whether the T-cells are fully activated. Can the authors provide additional evidence that this is the case, for example by showing the modulation of intracellular signaling molecules like Lck? I also suggest to include confidence intervals to the data shown in Fig. 2B.

4. I feel that the paper could be much more appealing to the average reader by improving the data presentation. Fig. 1 is a schematic, Fig. 2 shows a control experiment, Fig. 3A is also a control experiment, and the stoichiometry and probability density plots in Fig. 3B seem largely redundant. The highlight of the study is Fig. 4 and this should be featured more prominently. I find the analysis in Supp. Fig. 5 quite interesting, maybe parts of this can be featured in the main figure.

The estimation of the loading rates in Fig. 5 are also interesting, but plotting representative single-molecule traces seems somewhat anecdotal and is not very convincing. The figure legend of Fig. 5 indicates that the authors have recorded a great number of trajectories and I strongly encourage the authors to show these data. The single-molecule data is what distinguishes their study from previous work and it should be featured more prominently. I also wonder whether the experimentally observed FRET efficiencies can be mapped to the spatio-temporal single-molecule trajectories. Would that not be interesting to show?

I hope that the authors will be able to address these issues, which would then justify, in my opinion, the publication in Nature Communications.

Reviewer #2 (Remarks to the Author):

Goehring et al report a new FRET-based sensor equipped with a TCR-reactive single chain antibody fragment allowing the systematic dissection of mechanical force at the single molecule level. I feel very excited about this work because the authors present for the first time carefully executed force probing experiments of mechanical force at the molecular level. These experiments demonstrate that the TCR-pMHC interactions might be much more sensitive to physical force as previously thought and as such provide a very important contribution to the field. At the outset, I believe that it is important to state that there has been a myriad of force measurement studies in the recent years but the present one exceeds the quality and completeness from the biophysics and biology point of view.

Moreover I would like to state that I truly enjoyed reading the manuscript, which in my opinion was very well executed, structured, and technically sound presented. I would therefore recommend to accept the manuscript for publication in Nature Communications with minor corrections.

(1) I was wondering whether the authors might have mixed up the word kinetic and dynamical

behaviour in the abstract? The first sentence reads as "Mechanical forces acting on ligand-engaged T-cell receptors (TCRs) have previously been implicated in T-cell antigen recognition, yet their magnitude, spread, and dynamical behavior are still poorly defined." I guess either way could be justified with the results of the manuscript.

(2) Could the authors please add a statement to the "Statistics section" in how many cells the respective experiments have been performed?

(3) I think that the manuscript would benefit from the authors speculating about the implications of their findings in the discussion section regarding initial-triggering of single TCR-pMHC and later multi-receptor interactions within clusters during IS formation.

(4) In light of this, in the opinion of the authors, how do their findings fit into the context of actin protrusions facilitating T-cell activation such as actin ruffles during T-cell triggering. In addition, during IS formation, invadosomes and actin foci might generate active pushing and pulling forces but actin retrograde flow in the lamellipodium constant mechanical load.

Reviewer #3 (Remarks to the Author):

"Temporal Analysis of T-Cell Receptor-Imposed Forces via Quantitative Single Molecule Fret Measurements" by Göhring et al.

This paper by Göhring et al. describes analysis of forces applied by T cells to individual TCRs using a single molecule FRET based force sensor. The study compares TCR binding on fluid and gel supported lipid bilayers (SLBs). They execute both ensemble measurements and measurements where single molecule forces can be resolved through their H57 Fab linked tension sensor.

Unfortunately, there are a number of major caveats with how the experiment was set up that are coupled to interpretation of the work. These needs to be clarified first and will impact many of the conclusions of the paper. There are concerns regarding 1) binding strategy employed here compared to native, natural binding, 2) ligand number/concentration density in this study compared to digital triggering requirements 3) force interpretation (direction and magnitude), and 4) time zero and temporal progression with respect to interpreting these results with respect to initial signaling events vs. subsequent processes such as immunological synapse formation etc.

The force sensor is connected to the Fab of an antibody H57 that binds TCR beta but also is known to block activation of T cells (see Kim et al, 2009 JBC). H57 is a good marker for TCR beta. I don't believe that activation through H57 is a normal pathway and appropriate for drawing conclusions as stated in their work. Native binding is through pMHC, which binds the top of the variable domains of abTCR. H57 Fab binding is lower, near the FG loop and blocks conformational change and likely blocks CD3 ϵ binding. Force at this location would couple to the system differently than force applied through the top. In addition to blocking a conformational change that is linked to T cell activation, mechanotransduction could be altered. I do think that occasionally cells in the presence of H57 can activate. In rare cases a pMHC bound to a TCR in the presence of H57 can undergo a conformational change and activate a cell. This is reported in Feng et al, 2017 PNAS requiring 23 pN. Force through a single antibody (true digital) has also been shown in Feng figure 2C and requires 10pN. That said, it is interesting that some activation is seen in this work through direct force on

H57.

With respect to direct activation through H57, this appears to be force dependent. There is a recent work by Lin et al in Biophysical Journal 2020 (Groves Lab) that is relevant to this study. Here activation through an H57 FAB is seen but it requires many molecules and is not digital. Cell morphology is also altered.

The authors conclude “Of particular importance, we obtained evidence that molecular forces beyond 2pN exerted via the TCR are neither a consequence of nor a precondition for T cell signaling: ...”

The authors do note that T cell activation is digital, requiring one molecule. I do not think however they are able to observe this initial activating event. Measurements of Figure 2 are ensemble and done at high densities where such a rare digital activation event would be masked. It would not be detected in the population signal observed. The authors also seem to acknowledge this point in the first paragraph of “forces in the immunological synapse.” section on page 9. There are no measurements to support this conclusion at very low concentration such as seen in Feng.

Likewise, measurements of Figure 3 are able to observe single molecules but in the presence of many more unlabeled molecules. In this measurement signals are only seen by diluting unlabeled sensors at 50-100/ μm^2 with labeled sensors at $< 0.01 /\mu\text{m}^2$. 3-4 orders of magnitude lower concentration. Thus, a rare TCR digitally activating the cell among the unlabeled sensors would not show up in the single molecule observable. When they try these experiments at ligand densities where force based triggering has been achieved, no activation is observed. The related work by Lin noted above also required multiple molecules to activate, however they were able to see activation with H57 Fab at concentrations comparable to ligand density (at 0.2/ μm^2). It would be interesting to know if authors tested concentrations in between what is presented here, whether there was activation, and what were corresponding forces.

An additional problem is whether time zero can be established such that activation claims would be substantiated here and if not whether results should be interpreted as activation or more downstream such as synapse formation etc. In other works time zero is well defined such as Bio Force Probe (BFP) of Zhu or optical tweezers measurements where force activation has been quantified and is triggered actively with a high degree of temporal constraining. In this work, activation is initiated by dropping cells on lipid bilayers. It would be very hard to pick out initial triggering. Thus, it is likely that what is being observed in this work relates to post-activation events tied to rearrangements associated with immunological synapse formation. I believe a more plausible explanation for these measurements might be active transport of TCRs after activation occurs. This is seen in figure 5 of Feng et al, PNAS 2017 through a different antibody following activation. I believe this current work may actually measuring force exerted on the TCR's via actomyosin streaming after activation occurs somewhere else in time and space. It may be that the authors have nice evidence that streaming is single motor based. If it were coupled directly to an actin filament, force magnitudes through multiple motors might be much higher. The force measured here is low, consistent with the 3.8 pN stall force measured in Feng.

The paper has a nice comparison between gel phase and free lipids. It corroborates previous findings that force is primarily applied tangent to the cell surface through an elegant smFRET force sensor, however the interpretation should be more carefully considered.

Figure details:

- Fig 1A: both fluorophores are labeled “acceptor”
- Fig 2A and Supp Fig 2C: Scale bar not defined. What type of image is the top row?
- Supplementary Fig 2C: Scale bar not defined

Detailed response by the authors to the reviewers' comments (NCOMMS-20-15899)

“Temporal Analysis of T-Cell Receptor-Imposed Forces via Quantitative Single Molecule Fret Measurements” by Göhring et al.

We are grateful to the reviewers for the thorough reading of our manuscript and the overall positive assessment and constructive criticism. In the revised version we have addressed the reviewers' concerns, as is detailed in the point-by-point reply attached below. In particular, the following major amendments were made:

- Upon request of Reviewer #1 and #3, we developed a novel force sensor to quantify pulling forces exerted on IEk/MCC, the nominal / physiological ligand of the 5c.c7 T cell antigen receptor employed in our study. The new data are displayed as **Fig. 4**
- As was requested by Reviewer #1, we have added experiments showing the role of the actin cytoskeleton for the generation of pulling forces. The data are shown in new **Supplementary Fig. 6A**.
- Upon request by Reviewer #1 we have added data on CD8⁺ T-cells in new **Supplementary Fig. 6B**
- As requested by Reviewer #1 and #3, a more detailed assessment of T-cell activation was performed, including new analyses on intracellular calcium and also of synaptic recruitment of ZAP70 as an indicator of TCR-proximal signaling. The new data are shown in **Supplementary Fig. 1** and **Supplementary Fig. 3**.
- Upon request of all three Reviewers we have extensively modified the discussion section.
- Upon request by Reviewer #1, we streamlined the presentation in **Figs. 1 – 4** and shifted some information from the main figures to supplementary information.

Reviewer #1

Reviewer's Summary: This manuscript describes the development and application of a molecular tension sensing technique that allows the quantification of mechanical force across the T-cell receptor (TCR) in a single-molecule resolved fashion. The authors functionalized supported lipid bilayers (SLB) with a FRET-based tension sensor carrying a TCR-reactive single chain antibody. They seeded 5c.c7-derived T-cells onto these surfaces, and then started by performing FRET ensemble measurements. These experiments demonstrate the functionality of the approach indicating pN-scale forces across the T-cell receptor on gel-like SLB that allow for shear- and pulling forces. By contrast, forces are not observed on fluid-like SLB that are characterized by high lateral mobilities and do not allow the generation of tangential forces.

Next, the authors dilute the amount of tension sensor molecules in the SLB to perform single-molecule resolved force measurements. These experiments confirm the previous observations (i.e. tension on gel-like SLB but no or little force on fluid-like SLB) and reveal forces of about 4–5 piconewton (pN) per TCR-antibody linkage on gel-like SLB. *An unexpected finding is that forces appear to be largely independent of the T-cell activation state.* Finally, the authors perform a temporal analysis of the single-molecule FRET trajectories to determine an average loading rate across the TCR-linkage of about 1.5 pN/s.

Overall, this is a technically interesting study. Even though FRET-based tension sensors have already been applied to investigate molecular forces with single-molecule resolution in cells (Morimatsu et al 2013, Nano Lett), single-molecule-resolved force measurements have (to my knowledge) not been realized with molecular force sensors in T-cells before, which makes this study quite intriguing.

Response: We would like to thank the reviewer for his/her positive assessment of our manuscript.

Reviewer's Concern: There is no doubt that the physiological background is highly relevant, but the degree of biological insight is quite limited at this point, in part because a study using a similar force sensing approach to study T-cell mechanics in ensemble measurements has been published before (Liu et al, 2016, PNAS).

Response: We agree with Reviewer #1 that the pioneering paper by Liu et al. reports highly relevant results (Liu et al., 2016). However, the usage of a DNA-based force sensor comes with a number of disadvantages, which render the obtained results qualitatively and quantitatively different from ours:

- **Sensor nature:** The sensor was conceptually different from ours. The DNA-based molecular force sensor used in Liu et al. can only adapt two different FRET states (folded or unfolded) and is hence digital in design, whereas our peptide-based sensor inherently produces an analog readout with a dynamic range of 0.9 to 10 pN.
- **Kinetics and conversion into a force metric:** As acknowledged by Reviewer #1 in “major point 2”, we have discussed in our manuscript a main limitation of the DNA-based force sensor used by

Liu et al., which is the dependence of its unfolding probability on the loading rate. For example, if an average constant load of only a few pN were to be applied over a prolonged time, all sensors would eventually open up. Hence, it is *per se difficult to obtain a quantitative result with a sensor based on DNA unzipping*.

- **Sensor size:** The size of the sensor reported by Liu et al. is *rather large* (~36 nm for the composite of a 9nm gold particle, a double-stranded DNA of 48 bp, the ssDNA linker, a streptavidin and a pMHC molecule) when compared to the size of the synaptic gap (~14 nm, interface between target cell and T-cell). Our sensor features a size of ~12nm in the collapsed form.
- **Sensor platform:** All previous studies focused on molecular forces resulting from the use of immobile rigid surfaces such as glass and gold, with the exception of Ma et al. (Ma et al., 2016). In the latter study, force probe-decorated gold particles were tethered to fluid lipid bilayers. Although the gold particles were mobile, the force probes themselves were still immobilized on the rigid gold surface. As a consequence, multiple DNA tension probes and hence multiple TCR molecules were cross-linked via the solid gold nanoparticles, allowing for tensile forces between the cross-linked TCRs. In contrast, our paper reports for thirist time *a lipid-anchored force sensor. It was precisely because of working with this design that we were able to observe the marked difference in tensile forces exerted on rigid versus fluid surfaces*.
- **Single Molecule versus Ensemble Measurements:** The design reported by Liu et al. does not allow for single molecule measurements. Hence, neither force distributions nor the dynamics of the forces applied by the T-cell could be studied, as both were masked in the ensemble average.

We have now mentioned differences between the two approaches in both the introduction and have discussed them in more detail in the discussion section in the revised version of the ms.

Reviewer's Concern: More importantly, the here established approach seems to provide only limited insights into the mechanics of *physiologically occurring TCR-pMHC interactions*. Mechanisms of regulation by *co-receptors or cytoskeletal networks are also not investigated*, and important external factors that are thought to modulate T-cell mechanics (e.g. shear flow) are not considered.

Response: A large body of experimental have been added to the ms., which considerably increases the physiological relevance of our study. We address the reviewer's concerns in more detail in the "major points" section.

Reviewer's Suggestion: *I also feel that the T-cell activation experiments should be evaluated/controlled more rigorously*, and the authors should improve the way the data are presented. I would like to encourage the authors to address these issues and thereby increase the impact of this study. For more detailed comments, please see below.

Response: We complied with Reviewer #1's request to monitor the state of T-cell activation in additional experiments: in addition to imaging the rise of intracellular calcium we have quantitated in the revised version of the ms. synaptic ZAP70-recruitment as a direct indicator of TCR-proximal signaling (see Supplementary Figure 2).

Major point 1: Owing to the nature of the here established setup, the measurements do not probe a naturally occurring interaction between the TCR and the pMHC, but an artificial linkage mediated by the TCR and a single-chain antibody (H57-scFv). It is therefore unclear to which degree the here measured forces reflect the physiological situation, where highly specific pMHC–TCR interactions trigger the T-cell response. This seems to inherently limit the study, especially because the central hypothesis in the field of T-cell mechanics, that agonist-bound MHC-TCR interactions form catch-bond-like linkages (Liu et al, 2014, Cell), cannot be directly tested. I am not sure whether the authors can find a way to improve the specificity of their setup, but it would significantly increase the impact of this paper if more physiologically relevant measurements were possible.

Response: We agree with Reviewer #1 and have therefore included in the current version of the ms. experiments involving a force sensor featuring a peptide-loaded MHCII, more specifically IEk/MCC as the nominal ligand for the 5c.c7 TCR. The new data are shown in Figure 4, and the results are described on page 15.

Experiments were performed on gel-phase SLBs under conditions supporting either T-cell activation or antigen scanning. Under activating conditions, we observed a clearly visible shoulder towards high forces in the histogram, which corresponds to a force peak with a mean value of 2.0 ± 0.0 pN. This shoulder indicates the presence of pulling forces exerted onto the TCR-pMHC bond. Interestingly and in contrast to the scenario involving the use of MFS-H57, most single molecule signals showed rather low tensile forces close to the detection limit of 0.9 pN.

Taken together, experiments involving the use of the natural ligand confirm the presence of pulling forces also for physiologically relevant antigen recognition events. The smaller force magnitude suggests, however, that the TCR-pMHC interaction is of slip-bond type, which would be in agreement with our previous results (Huppa et al., 2010). Of note, our new data do not completely rule out the presence of catch bonds. In that case, however, catch bonds would need to show a transition to slip bonds at rather low forces below ~ 1 pN.

We provided an extensive discussion of the new data on page 17.

Major point 2: The authors have acknowledged that previous work has been published, which is conceptually quite similar to the here used approach (Liu et al, 2016, PNAS). I do not think that this is a problem per se, because this study here addresses T-cell mechanics on a different scale (i.e. single-molecule resolution). I also agree with the authors that the loading rate sensitivity of DNA-based tension sensors is quite problematic when it comes to the calculation of molecular forces. Nevertheless, the authors should

provide much more biological experiments to document the advantages of the here established technique.

Such experiments could include the use of different T-cell subtypes, exploring effects of distinct T-cell co-receptors (e.g. CD8/CD8, ICAM, integrins), or the application of shear flow. The authors should also use inhibitors to test whether the observed forces across the TCR on gel-like substrates are subject to cytoskeletal regulation (Thauland et al. 2017, Sci Signal).

Response: We thank the reviewer for acknowledging the novelty of our approach, and for suggesting experiments to strengthen the impact of our study. We would like to point out here, that such single molecule experiments are time-consuming, as each immunological synapse provides only a few trajectories. Hence, multiple weeks of experiments are required to obtain single force histograms. We hence tried to prioritize our new experiments such that they cover as many of the suggested aspects as possible. The following new experimental settings were included in the revised manuscript:

- i) We were able to construct a new force sensor, in which the spring element is linked to the natural ligand pMHC (see our answer to Major Point 1). Since the magnitude of forces with the natural ligand was close to our detection limit, we decided that the influence of coreceptors would be difficult to assess at this stage: a different force sensor optimized for sensitivity at low pulling forces would be required to perform such experiments at sufficient quality.
- ii) We performed additional experiments using the primary CD8⁺OT-1-TCR-transgenic T-cells investigating the TCR-exerted forces with the scFv H57-coupled force sensor. The new data are included as new **Supplementary Fig. 6B**, and described in the text on **page 13**. Briefly, we observed similar histograms as for the CD4⁺5c.c7-transgenic T-cells, yielding a high force peak at 6.1 ± 0.1 pN and 5.2 ± 0.1 pN for activating and scanning conditions, respectively.
- iii) We investigated the role of the actin cytoskeleton in force application by inhibiting actin polymerization via cytochalasin D. As expected, we observed nearly complete abrogation of pulling forces. These data are included as new **Supplementary Fig. 6A**, and described in the text on **page 11**.

While we consider the role of shear forces for antigen recognition as an important topic *per se*, our force sensor measurements could hardly contribute new insights at this stage: it has been shown that shear stress increases the forces acting on the TCR-pMHC bond depending on the magnitude of the shear force (see e.g. (Limozin et al., 2019)). Such an experiment hence would have essentially provided a positive control for our force sensor. As we are confident that the control experiments sufficiently demonstrate the performance of our constructs, we ranked the shear force experiments as less relevant for this manuscript. In the future, it will indeed be interesting to study the influence of shear force on the loading rate applied to the MFS-IE^k/MCC.

Major point 3: An intriguing finding of this study is that very small forces support T-cell stimulation. A critical data set for this interpretation is Supp. Fig. 2B showing data from ratiometric calcium imaging. These

data sets demonstrate that adding molecular force sensors at sufficiently high concentrations to SLB can induce a Ca²⁺ response. However, the response is clearly different from the IE^k-MCC-induced condition, which leads to a quicker (gel-like SLB) or more potent (fluid-like SLB) response. *The question therefore is whether the T-cells are fully activated. Can the authors provide additional evidence* that this is the case, for example by showing the modulation of intracellular signaling molecules like Lck? I also suggest to include confidence intervals to the data shown in Fig. 2B.

Response: We agree with Reviewer #1's assessment and investigated T-cell activation in more detail. We provide now *additional analyses of the calcium signaling in Supplementary Fig. 1F, G and Supplementary Fig. 3A-C*, which includes in particular the interquartile ranges for the different constructs and conditions, and a peak amplitude analysis of the synchronized T-cell activation profiles. Briefly, the peak calcium signals were identical with the positive control, i.e. direct stimulation via SLB-anchored pMHC, and sustained over at least 15 minutes. As correctly spotted by Reviewer #1, T-cell activation was slightly delayed compared to the positive control, possibly reflecting a slightly altered mechanism by which TCR-engagement with the force sensor constructs triggers activation.

In addition, we followed the request of the reviewer to further analyze downstream signaling. We opted for the analysis of ZAP70-recruitment upon seeding cells gel-phase or fluid-phase SLBs decorated with MFS-H57 (**Supplementary Fig. 3D**). For both SLBs we observed ZAP70 recruitment under activating conditions, whereas no recruitment could be detected under scanning conditions. This further confirms our classification of the single molecule experiments into “scanning condition” and “activated condition”.

Major point 4: I feel that the paper could be much more appealing to the average reader *by improving the data presentation*. Fig. 1 is a schematic, Fig. 2 shows a control experiment, Fig. 3A is also a control experiment, and the stoichiometry and probability density plots in Fig. 3B seem largely redundant. The highlight of the study is Fig. 4 and this should be featured more prominently. I find the analysis in Supp. Fig. 5 quite interesting, maybe parts of this can be featured in the main figure.

The estimation of the loading rates in Fig. 5 are also interesting, but plotting representative single-molecule traces seems somewhat anecdotal and is not very convincing. The figure legend of Fig. 5 indicates that the authors have recorded a great number of trajectories and I strongly encourage the authors to show these data. The single-molecule data is what distinguishes their study from previous work and it should be featured more prominently. I also wonder whether the *experimentally observed FRET efficiencies can be mapped to the spatio-temporal single-molecule trajectories*. Would that not be interesting to show?

Response: We thank the reviewer for the suggestions regarding the data presentation. We have reorganized the flow of the figures:

- The new **Fig. 1** contains in addition to the schematic also the proof-of-principle ensemble experiment, and representative single molecule FRET images.

- The FRET histograms of previous Fig. 3B have been moved to new **Supplementary Fig. 4** to reduce redundancy.
- **Fig. 2** of the revised ms. contains the single molecule force histograms (previous Fig. 4)
- **Fig. 3** of the revised version contains the analysis of single molecule force trajectories. We realized that the description in the original ms. lacked clarity, and we therefore modified the text: in fact, the data set shown in subpanel c is an average over those seven trajectories, which showed clearly identifiable force ramps.
- We introduced a new **Fig. 4**, which contains the new data set on the natural 5c.c7 TCR ligand MFS-IE^k/MCC.
- **Supplementary Fig. 6A** now features the cytoskeletal manipulation data.
- We decided to keep the previous Supplementary Fig. 5 (now **Supplementary Fig. 8**) in the supplements, as it represents only a control experiment for the mobility and specificity of the MFS-H57, and appeared less essential to convey the overall message of our manuscript.
- **Supplementary Fig. 6B** now features results concerning OT1 TCR-transgenic CD8⁺ T-cells interacting SLB functionalized with MFS-H57.
- **Supplementary Figs. 9 and 10** now also display new spatiotemporal analyses of the trajectories. For this, we pooled data with respect to the elapsed time from first contact between the T-cell and the functionalized SLB in groups of 5 minutes. The results revealed remarkable differences between scanning and activated T-cells. For activated T-cells (new **Supplementary Fig. 9A**), we observed the emergence of the high force peak only after ~10 minutes at 5.6 pN, where it stayed over the remaining observation time. Tensile forces occurred both in the cell periphery as well as in central regions of the immunological synapse (new **Supplementary Fig. 10A**). In contrast, scanning T-cells exhibited the high force peak immediately upon first SLB contact (new **Supplementary Fig. 9B**). Interestingly, the average pulling force declined with time from 7.5 pN in the phase of early contact, to 6.3 pN between 5 and 10 minutes, down to low force magnitudes below 3 pN after 10 minutes. Again, we did not observe any spatial preference of forces (**Supplementary Fig. 10B**).

Reviewer's Conclusion: I hope that the authors will be able to address these issues, which would then justify, in my opinion, the publication in Nature Communications.
--

Response: We thank Reviewer #1 for his/her excellent suggestions. We hope that he/she finds our manuscript of sufficient interest and impact for publication in *Nature Communications*.

Reviewer #2

Reviewer's Summary: Goehring et al report a new FRET-based sensor equipped with a TCR-reactive single chain antibody fragment allowing the systematic dissection of mechanical force at the single molecule level. I feel very excited about this work because the authors present for the first time carefully executed force probing experiments of mechanical force at the molecular level. These experiments demonstrate that the TCR-pMHC interactions might be much more sensitive to physical force as previously thought and as such provide a very important contribution to the field. At the outset, I believe that it is important to state that there has been a myriad of force measurement studies in the recent years but the present one exceeds the quality and completeness from the biophysics and biology point of view. Moreover I would like to state that I truly enjoyed reading the manuscript, which in my opinion was very well executed, structured, and technically sound presented. I would therefore recommend to accept the manuscript for publication in Nature Communications with minor corrections.

Response: We thank the reviewer for his/her very positive assessment of our ms..

(1) I was wondering whether the authors might have mixed up the word kinetic and dynamical behaviour in the abstract? The first sentence reads as "Mechanical forces acting on ligand-engaged T-cell receptors (TCRs) have previously been implicated in T-cell antigen recognition, yet their magnitude, spread, and dynamical behavior are still poorly defined." I guess either way could be justified with the results of the manuscript.

Response: We agree with Reviewer #2 and thank him/her for pointing this out. We wish to indicate the time-dependence, and hence used the term "temporal" instead.

(2) Could the authors please add a statement to the "Statistics section" in how many cells the respective experiments have been performed?

Response: In the figure legends, we provide the number of the analyzed movies contributing to the corresponding data set, where each movie comprises between 1 and 4 T-cells. This sentence was added now to the subsection "Statistical analysis" of the Materials and Methods section. In addition, we added a clarifying statement to the legend of **Fig. 2**.

(3) I think that the manuscript would benefit from the authors speculating about the implications of their findings in the discussion session regarding initial-triggering of single TCR-pMHC and later multi-receptor interactions within clusters during IS formation.

(4) In light of this, in the opinion of the authors, how do their findings fit into the context of actin protrusions facilitating T-cell activation such as actin ruffles during T -cell triggering. In addition, during IS formation, invadosomes and actin foci might generate active pushing and pulling forces but actin retrograde flow in the lamellipodium constant mechanical load.

Response: We thank the reviewer for these suggestions and added an extensive discussion on **page 17 to 19** of the revised ms..

Reviewer #3

Reviewer's Summary: This paper by Göhring et al. describes analysis of forces applied by T-cells to individual TCRs using a single molecule FRET based force sensor. The study compares TCR binding on fluid and gel supported lipid bilayers (SLBs). They execute both ensemble measurements and measurements where single molecule forces can be resolved through their H57 Fab linked tension sensor.

Reviewer's Concerns: Unfortunately, there are a number of major caveats with how the experiment was set up that are coupled to interpretation of the work. These needs to be clarified first and will impact many of the conclusions of the paper. There are concerns regarding 1) binding strategy employed here compared to native, natural binding, 2) ligand number/concentration density in this study compared to digital triggering requirements 3) force interpretation (direction and magnitude), and 4) time zero and temporal progression with respect to interpreting these results with respect to initial signaling events vs. subsequent processes such as immunological synapse formation etc.

Response: We thank Reviewer #3 for his/her constructive criticism of our study. We are confident we have addressed all points raised (for details see below under "major points").

Reviewer's Major Points: The force sensor is connected to the Fab of an antibody H57 that binds TCR beta but also is *known to block activation of T cells* (see Kim et al, 2009 JBC). H57 is a good marker for TCR beta. I don't believe that activation through H57 is a normal pathway and appropriate for drawing conclusions as stated in their work. Native binding is through pMHC, which binds the top of the variable domains of abTCR. H57 Fab binding is lower, near the FG loop and blocks conformational change and likely blocks CD3 ϵ binding. Force at this location would couple to the system differently than force applied through the top. In addition to blocking a conformational change that is linked to activation, *mechanotransduction could be altered*. I do think that occasionally cells in the presence of H57 can activate. In rare cases a pMHC bound to a TCR in the presence of H57 can undergo a conformational change and activate a cell. This is reported in Feng et al, 2017 PNAS *requiring 23 pN*. Force through a single antibody (true digital) has also been shown in Feng figure 2C and requires 10pN. That said, *it is interesting that some activation is seen in this work through direct force on H57*. With respect to direct activation through H57, this appears to be force dependent. There is a recent work by Lin et al in Biophysical Journal 2020 (Groves Lab) that is relevant to this study. Here activation through an H57 FAB is seen but it requires many molecules and is not digital. Cell morphology is also altered.

Response: We thank Reviewer #3 for critically examining the mechanistic implications of our work. We agree that the quantification of tensile forces acting on the natural ligand – peptide-loaded MHC – is important and highly relevant also in the context of our ms.. We hence present in the revised version of the ms. experiments involving the use of a force sensor, in which the spring unit was coupled to IE^k/MCC

(termed MFS- IE^k/MCC), the nominal ligand of the 5c.c7 TCR, for which T-cells employed in our study are transgenic (*see Fig. 4 with the results being described on page 15*).

Experiments were performed on gel-phase SLBs, both under activating and scanning conditions. Under activating conditions, we observed a clearly visible shoulder towards high forces in the histogram, which corresponds to a force peak with a mean value of 2.0 ± 0.0 pN. This shoulder shows the presence of pulling forces exerted onto the TCR-pMHC bond.

We would like point out, that the primary motivation behind this study was not to scrutinize the biomechanical force sensor model of the TCR. Instead, we aimed here at providing a first molecular quantification of pulling forces exerted by T-cells via their TCR, as such numbers are currently not known. To justify our take on TCR-imposed forces we would like to lay out in the following shortcomings of previously published approaches towards force quantitation:

- Salaita and coworkers pioneered the use of DNA hairpin-based tension sensors (Liu et al., 2016). In this sensor design, tensile forces lead to the unzipping of a DNA hairpin, thereby changing the distance between a fluorophore and a quencher. We have already discussed some implications of such sensor constructs in the introduction section of our paper (**page 4**). For example, it is known that both the loading rate (Cocco et al., 2001) and the duration of the applied force (Mosayebi et al., 2015) affect the force required to unzip hairpins. Both constitute *a priori* unknown parameters. If given sufficient time, all DNA duplexes will eventually separate, even if very weak forces are applied. In contrast, the flagelliform sensor described in our manuscript provides a direct and quantitative readout of the applied forces, irrespective of their duration and loading rate.
- In a follow-up paper, the same group reported forces amounting to up to 4.7 pN for T-cells that interact with planar glass-supported lipid bilayers (SLBs) featuring pMHCs anchored to gold particles (Ma et al., 2016). However, in that study multiple DNA tension probes, and, as a consequence, multiple TCR molecules were cross-linked via the solid gold nanoparticles, allowing for tensile forces between the cross-linked TCRs. Also this aspect is discussed in our paper (**page 16**).
- Lang, Reinherz and coworkers presented a different approach, which was not based on the measurement of forces exerted by T-cells, but in contrast was set up to address whether forces *per se* trigger the activation of T-cells (Feng et al., 2017). They indeed found that low doses of agonistic peptide presented via MHC suffice to trigger Ca²⁺ response, as long as pMHC was pulled tangentially to the T-cell surface. The pulling was achieved using pMHC-coated beads trapped in a laser-optical tweezer. Forces were quantified by means of the drag force applied to the bead via the optical tweezer. This value, however, contains contributions from multiple TCR-pMHC bonds (as pointed out by the authors e.g. in Fig. S8 (Feng et al., 2017)). In order to provide a quantification of the forces exerted at the level of single bonds, Feng et al. estimated the number of TCR-pMHC bonds formed between the bead and the T-cell from the average pMHC density on the bead surface. This procedure, however, appears problematic: binding probability does not only depend on the

pMHC density on the bead surface, but also on the elastic properties of the linkers, and the deformation of the T-cell membrane next to the bead, which likely leads to a partial engulfing of the bead. This complicates the discrimination between tangential and perpendicular forces. Even more so, pMHC is not homogeneously distributed over the bead surface; rotation of the bead in the optical trap will eventually orient the bead to maximize the number of pMHC-TCR interactions.

Reviewer's Concern: The authors conclude “Of particular importance, we obtained evidence that molecular forces beyond 2pN exerted via the TCR are neither a consequence of nor a precondition for T cell signaling: ...”

The authors do note that T cell activation is digital, requiring one molecule. I do not think however they are able to observe this initial activating event. Measurements of Figure 2 are ensemble and done at high densities where such a rare digital activation event would be masked. It would not be detected in the population signal observed. The authors also seem to acknowledge this point in the first paragraph of “forces in the immunological synapse.” section on page 9. There are no measurements to support this conclusion at very low concentration such as seen in Feng.

Likewise, measurements of Figure 3 are able to observe single molecules but in the presence of many more unlabeled molecules. In this measurement signals are only seen by diluting unlabeled sensors at 50-100/ μm^2 with labeled sensors at $< 0.01 /\mu\text{m}^2$. 3-4 orders of magnitude lower concentration. ***Thus, a rare TCR digitally activating the cell among the unlabeled sensors would not show up in the single molecule observable.***

Response: We thank the Reviewer #3 for carefully scrutinizing our statements, and we partially agree with him/her on this point. Our statement, which is based on measurements presented in **Fig. 2** (formerly Fig. 4), contains two aspects:

- Low ligand densities of 3-4 MFS molecules per cell did not trigger calcium flux. Still, at gel-phase membranes we observed force peaks larger than 2 pN under scanning conditions. Hence, our statement that forces beyond 2 pN exerted via the TCR are not a consequence of T-cell signaling appears justified.
- Experiments under activating conditions conducted with the use of fluid lipid bilayers did not show any indication of a high force peak, although important hallmarks of early T-cell signaling were observable, which include TCR microcluster formation, calcium flux, as well as ZAP70 recruitment (new **Supplementary Fig. 1 F, G, and Supplementary Fig. 3**). We do not agree with Reviewer #3 that under such experimental conditions we may have missed those few binding events, which were responsible for T-cell activation. This is primarily because TCR-triggering is required at all times to sustain TCR-proximal signaling, to maintain synapse integrity and to ensure the unfolding of the full effector potential of helper T-cells, as we have demonstrated in 2003 (Huppa et al., 2003). It is certainly possible, as we have now mentioned in the Discussion section, that we may have missed the earliest trigger events. However, we do not see this as a fundamental problem, since (i) all

synaptic TCR-pMHC binding events contribute to T-cell signaling (see above) and (ii) since we have recorded a considerable number of trigger events at very early (and also later) stages of synapse formation. Results have been categorized with regard to time after first bilayer contact (see **Supplementary Fig. 9**) and described on **page 12**. Nonetheless, as we explain in more detail below, we are highly motivated to measure and analyze the very first events, which is technically very demanding (and risky) and which is at current outside the scope of our study.

Reviewer's Concern: When they try these experiments at ligand densities where force-based triggering has been achieved, no activation is observed. The related work by Lin noted above also required multiple molecules to activate, however they were able to see activation with H57 Fab at concentrations comparable to ligand density (at $0.2/\mu\text{m}^2$). It would be interesting to know if authors tested concentrations in between what is presented here, whether there was activation, and what were corresponding forces.

Response: Indeed, we have also observed quantitative differences. As a matter of fact, our own dose-response curves for the H57 and 5c.c7 T-cells (Hellmeier et al., 2020) yielded ~10-fold higher ligand densities for half maximal T-cell triggering (as determined via the recorded calcium signal) compared to the data reported by Lin et al (Lin et al., 2020). Part of this difference may arise from using different T-cells (in our case 5c.c7 TCR-transgenic, in the case of Lin et al. AND TCR-transgenic T-cells). Indeed, comparative experiments between the two TCRs in our lab showed a lower activation threshold for AND TCR-transgenic T-cells when compared to 5c.c7 TCR-transgenic T-cells, which may reflect differences in H57 scFV-TCR β binding (as there appear to be structural variations in the H57 epitope, which are TCR-dependent) or alternatively a slightly altered wiring of the underlying signaling machinery. After all, the highly artificial AND TCR, which arose after mispairing α and β TCR subunits, interacts with its nominal ligand IE^k/MCC with non-physiological binding constants (extremely low off-rates). Of note, transgenic mice that are homozygous for the AND TCR and are crossed on a B10.BR background (i.e. they express two copies of IE^k and two copies of the AND TCR) do not give rise to any mature T-cells in the periphery due to negative selection.

We agree with Reviewer #3 that it would be an ultimate goal to study forces for those TCR-pMHC interactions that actually provide the initial activation trigger. Experimentally, however, this is extremely difficult. While the bead assay employed by Feng et al. (Feng et al., 2017) allows for high temporal definition, single molecule observations in such a setting are difficult if not impossible. Conversely, our approach provides single molecule sensitivity at the expense of temporal resolution. Unfortunately, also the suggested experiments involving intermediate densities would hardly provide the window to initial triggering events: in practice, even if no dilution of the MFS with unlabeled MFS was required, still a large fraction of signals could not be included in the analysis due to ambiguities arising from photobleaching of FRET donor or acceptor (see the description of our filtering processes on **page 29 to 30**). We would like to emphasize at this point that quantifying single molecule FRET of mobile molecules in a live cell setting is

still considered very challenging. After having published the preprint of this ms. on bioRxiv, multiple research groups have already asked us to provide the data analysis platform, as it is currently one of the very few platforms that robustly facilitates such analysis.

Regardless, we believe we will be able to tackle this problem in the future e.g. by placing single T-cells onto the activating surfaces with high temporal definition e.g. with the use of microfluidic devices; such experiments, however, go beyond the scope of our current paper. At this stage, we were able to group experiments according to the time point when T-cells formed contact with the SLB (see our answer to the next question).

Reviewer's Concern: An additional problem is whether time zero can be established such that activation claims would be substantiated here and if not whether results should be interpreted as activation or more downstream such as synapse formation etc. In other words time zero is well defined such as Bio Force Probe (BFP) of Zhu or optical tweezers measurements where force activation has been quantified and is triggered actively with a high degree of temporal constraining. In this work, activation is initiated by dropping cells on lipid bilayers. It would be very hard to pick out initial triggering. Thus, it is likely that what is being observed in this work relates to post-activation events tied to rearrangements associated with immunological synapse formation. I believe a more plausible explanation for these measurements might be active transport of TCRs after activation occurs. This is seen in figure 5 of Feng et al, PNAS 2017 through a different antibody following activation. I believe this current work may actually measuring force exerted on the TCR's via actomyosin streaming after activation occurs somewhere else in time and space. It may be that the authors have nice evidence that streaming is single motor based. If it were coupled directly to an actin filament, force magnitudes through multiple motors might be much higher. The force measured here is low, consistent with the 3.8 pN stall force measured in Feng.

Response: The comment of the reviewer has prompted us to carry out new experiments, in which we precisely recorded the time-point at which T-cells contacted the SLBs. For each experimental run we established a global time-frame, with time-point zero referring to the first T-cell contact. The additional statistics allowed us to pool single molecule FRET data with respect to this global time. The results revealed astounding differences between scanning and activated T-cells. For activated T-cells, we observed the emergence of the high force peak only after ~10 minutes at 5.6 pN, where it stayed over the remaining observation time. In contrast, scanning T-cells exhibited the high force peak immediately upon contacting the SLB. Interestingly, the average pulling force declined with time from 7.5 pN in the phase of early contact, to 6.3 pN between 5 and 10 minutes, down to low force magnitudes below 3 pN after 10 minutes. The new data are included as **Supplementary Fig. 9** and described on **page 12**.

We further discussed these data on **page 18** in the discussion section. Briefly, the emergence of reduced pulling under scanning conditions may reflect an adaptation to the situation of low stimulus: for example, T-cells may well need to switch to a more sensitive scanning mechanism in order to detect weak stimuli.

The observed delay in the emergence of pulling forces in case of T-cells encountering activating gel-phase membranes supports the interpretation of forces mediated by microcluster transport, as microclusters need to be formed beforehand, which takes a few minutes.

The paper has a nice comparison between gel phase and free lipids. It corroborates previous findings that force is primarily applied tangent to the cell surface through an elegant smFRET force sensor, however the interpretation should be more carefully considered.

Response: We would like to thank Reviewer #3 for his/her overall positive assessment, and we are confident that our revised ms. now merits publication in *Nature Communications* given the large body of new data added and a more careful description and contextual discussion of the results.

Figure details:

- Fig 1A: both fluorophores are labeled “acceptor”
- Fig 2A and Supp Fig 2C: Scale bar not defined. What type of image is the top row?
- Supplementary Fig 2C: Scale bar not defined

Response: We thank Reviewer #3 for spotting these errors which we have corrected in the revised version of the ms..

References

- Cocco, S., R. Monasson, and J.F. Marko. 2001. Force and kinetic barriers to unzipping of the DNA double helix. *Proceedings of the National Academy of Sciences of the United States of America*. 98:8608-8613.
- Feng, Y., K.N. Brazin, E. Kobayashi, R.J. Mallis, E.L. Reinherz, and M.J. Lang. 2017. Mechanosensing drives acuity of $\alpha\beta$ T-cell recognition. *Proceedings of the National Academy of Sciences*. 114:E8204-E8213.
- Hellmeier, J., R. Platzer, A.S. Eklund, T. Schlichthärle, A. Karner, V. Motsch, E. Kurz, V. Bamieh, M. Brameshuber, J. Preiner, R. Jungmann, H. Stockinger, G.J. Schütz, J.B. Huppa, and E. Sevcsik. 2020. DNA origami demonstrate the unique stimulatory power of single pMHCs as T-cell antigens. *bioRxiv:2020.2006.2024.166850*.
- Huppa, J.B., M. Axmann, M.A. Mortelmaier, B.F. Lillemeier, E.W. Newell, M. Brameshuber, L.O. Klein, G.J. Schütz, and M.M. Davis. 2010. TCR-peptide-MHC interactions in situ show accelerated kinetics and increased affinity. *Nature*. 463:963-967.
- Huppa, J.B., M. Gleimer, C. Sumen, and M.M. Davis. 2003. Continuous T cell receptor signaling required for synapse maintenance and full effector potential. *Nat Immunol*. 4:749-755.
- Limozin, L., M. Bridge, P. Bongrand, O. Dushek, P.A. van der Merwe, and P. Robert. 2019. TCR-pMHC kinetics under force in a cell-free system show no intrinsic catch bond, but a minimal encounter duration before binding. *Proceedings of the National Academy of Sciences*. 116:16943-16948.
- Lin, J.J., G.P. O'Donoghue, K.B. Wilhelm, M.P. Coyle, S.T. Low-Nam, N.C. Fay, K.N. Alfieri, and J.T. Groves. 2020. Membrane Association Transforms an Inert Anti-TCR β Fab' Ligand into a Potent T Cell Receptor Agonist. *Biophysical Journal*. 118:2879-2893.
- Liu, Y., L. Blanchfield, V.P.-Y. Ma, R. Andargachew, K. Galior, Z. Liu, B. Evavold, and K. Salaita. 2016. DNA-based nanoparticle tension sensors reveal that T-cell receptors transmit defined pN forces to their antigens for enhanced fidelity. *Proc. Natl. Acad. Sci*. 113:5610-5615.
- Ma, V.P.-Y., Y. Liu, L. Blanchfield, H. Su, B.D. Evavold, and K. Salaita. 2016. Ratiometric tension probes for mapping receptor forces and clustering at intermembrane junctions. *Nano Letters*.
- Mosayebi, M., A.A. Louis, J.P. Doye, and T.E. Ouldridge. 2015. Force-Induced Rupture of a DNA Duplex: From Fundamentals to Force Sensors. *ACS nano*. 9:11993-12003.

Reviewer #1 (Remarks to the Author):

The revised manuscript by Göhring et al now includes a number of additional experiments that have further improved the manuscript. Since most of the reviewers' concerns have been adequately addressed (thanks for the very detailed rebuttal letter), I can fully support the publication of this interesting study. I only have a few suggestions:

1. I would like to encourage the authors to include more of the supplementary data into the manuscript. Fig.3 and Fig. 4 are quite slim, while Supplementary Figs. 9 and 10 show very interesting data sets. Would it not be worth featuring these supplementary data in one of the main figures?
2. The authors should check whether all references to the main figures are correct; the reference to Fig. 3B on page 14, for instance, is not fitting. (I think it should refer to Fig. 3C).

Reviewer #3 (Remarks to the Author):

"Temporal Analysis of T-Cell Receptor-Imposed Forces via Quantitative Single Molecule Fret Measurements" by Göhring et al.

The authors couple a fluorescence based force sensor to both H57 and natural ligands that bind the T cell receptor. The ligands are displayed on lipid bilayers that are supported where they can sustain a force or fluid where they move in two dimensions. Two densities of sensors are used ensemble 50-100 molecules/ μm^2 and low/single molecule sensor of 0.01/ μm^2 interspersed within dark constructs at higher ligand density 50-100/ μm^2 such that activating conditions can be achieved (see page 10). These activating conditions are balanced with studies at low density "scanning". There are lots of good results in the data and it would benefit the community to see this work.

While I really like the measurements and design of the experiments, I think I have a different model than the authors for interpreting their measurements. In some places the authors overreach with respect to how T cells are activated. Below are a few of the particular statements and conclusions where I do not see support from the measurements that should be modified or addressed before publication. Other points for consideration and suggestions are included as well.

In the abstract I do not understand the statement "recalibrate their significance in antigen recognition." Much of this work addresses forces on TCRs with and without activation conditions being present, not the actual antigen recognition event / triggering / activation itself. What exactly is the conclusion here about antigen recognition? I don't think the measurements are sensitive to these rare events. In their rebuttal to the first review the authors agreed that they do not have the temporal resolution to see early triggering events, however that is not expressed in the manuscript.

Point to consider to avoid confusion in the field: The statement on page 7 second paragraph: "It is known that monomeric membrane-bound H57 activates T-cells with similar activation thresholds as agonistic..." should be qualified by stating that in free solution or under no force conditions H57 blocks activation. A reader not familiar with H57 might not understand H57Fab generally blocks activation except under bound conditions where force can be loaded.

In the introduction, p5 last paragraph it states “perpendicular tensile forces smaller than 2pN are already effective during the initiation of TCR signaling.” One cannot rule out short transients where higher force is exerted or rare events that are not seen, particularly since the number of unlabeled sensors are 3-4 fold higher than the labeled sensors. I think the 2pN the authors use arises from the average force distribution seen. (Short transients/spikes in figure 3A are seen going well above the average observed force.) In other studies at high ligand density external forces are not required. Yet it is possible that under these conditions force events occur within the dense interface arising from internal cell forces, the subject of the studies here. So I think it is a good point for the authors to address if possible as I feel it is an open important question. If there are potential transients this should be mentioned.

A related point on Page 17 “(ii) Furthermore, our results render it unlikely that forces larger than 2 pN are imposed on the TCR are required to trigger the TCR...” Note that the only cases where the authors have triggered the TCR are for high ligand density, and T cells are known to trigger with as few as 1 or 2 molecules with force. This statement should be qualified for high ligand density.

A point to consider: On page 15 top paragraph in the discussion of the activating conditions and scanning conditions for -IEK/MCC the sentence “The latter finding may result from early TCR-pMHC bond rupture...” It is likely that the lower force observed in activating compared to H57 also is due to early rupture.

The statement “Our observation of reduced pulling forces in the case of MFS-IEK/MCC would hence be in agreement with the presence of slip bonds formed between...” Note that as long as the bond lifetime is shorter than TCR-H57 (as is the case for both slip bonds and catch bonds in TCRs) one would expect to observe a reduced pulling force if the force is applied through some ramping mechanism (motor based or actin based transport) where the bond will let go before it gets to full force leading towards a reduced apparent force magnitude. I don't see how these results relate to catch or slip bond discussions.

Discussion of slip bond and catch bond bottom of page 17/top of page 18. I think the catch bond only starts developing above 5 pN (See Das et al, PNAS 2015, Fig 4 and figs 2 and 3) I am not sure the speculation about the catch/slip bond is relevant here. Also see Hwang et al PNAS 2020 discussion of forces required to bring out catch bond organization.

Point for consideration: Discussion on page 18 regarding perpendicular pulling forces. As a cell lands on a SLB fluid surface, its cytoskeletal machinery is not necessarily organized relative to the plane of the coverglass. Thus with all of the invagination and local microstructure of the cell surface there is likely, especially early on, geometries of the bonding interface where TCRs can pull perpendicularly against the surface. As the cell flattens out however, the machinery gets organized largely more parallel to the surface where the sensors are no longer able to be pulled as opposed to dragged within the fluid SLB. I think this might be a better explanation for the elegant temporal progression results showing forces seen for landing cells.

The introduction includes some good background on TCR forces through methods such as micropillars, deformable surfaces and even DNA based sensors but does not mention a number of single molecule studies where forces are directly measured on TCRs and motors linked to TCRs. It would be good to consider including some of these relevant works.

P16 middle paragraph. The force required to pull the sensor is an estimate based on 6 lipids and 7 pN/lipid and a parallel anchor assumption. Based on the structure with 3 inactive sites it is not clear that this species will form parallel anchoring points in the lipid bilayer. I think it is necessary to verify the sensor can sustain higher force experimentally or cite a reference where the mSAV-3xHis_6 had been able to sustain loads well above the sensor range in a similar SLB. If the load is not shared evenly it could break below 42 pN. If the sensor breaks free from the lipid bilayer at ~7 pN this could be the reason no forces much above this value are observed.

Point to consider: Figure 2b plots track count vs. diffusion parameter. Some of the trajectories in supplementary Figure 8 look directed. Does the MSD slope show diffusion? For the mobile spots it would be interesting to know the velocity if they are directed. I am interested in this because my interpretation is that a number of these motions are actin-myosin based transport. How about plotting velocity as well as diffusion parameter?

For consideration: On page 13 temporal analysis of scanning t cells is interesting. This might reflect reorganization of actin-myosin microstructure as the cell settles down.

Point to consider: On page 14 how long in time were the trajectories of the low FRET traces? Individual motors have a limited processivity. It is likely that these more stable higher frame rate trajectories are a single event and the longer time trajectories 71 seconds include many different motor/actin scenarios which would explain the increase in spread of force distribution observed.

Discussion: Page 15 “a clutch-like coupling between cortical actin and microcluster-resident TCRs ...” Seems there is good evidence in this paper that it is motor.

The statement “Interestingly, our time resolved experiments did not reveal... Such force ramps not only affect the mean but distribution of bond lifetimes...” From figure 3C, the loading rate appears to be about 2 pN/s. This is a very low loading rate case. That same figure shows trajectories that are flat over similar time windows. The data in figure 3C was taken from 7 single-molecule force ramps. What percentage of the time within these trajectories (duty cycle) show a force ramp? At least half of the 50 FPS trace appears flat and the second half of the 10 FPS trace in figure 3A. If much of the duty cycle is flat it can't be claimed that there is always a force ramp. I suspect many of the motors are off or under stalled conditions.

Discussion page 19 about cytoskeletal drag, actin flow rates etc. It seems to me the results here are very consistent with motor-based transport of TCRs following activation. This was pointed out in the earlier review:

From before: “I believe this current work may actually measuring force exerted on the TCR's via actomyosin streaming after activation occurs somewhere else in time and space. It may be that the authors have nice evidence that streaming is single motor based. If it were coupled directly to an actin filament, force magnitudes through multiple motors might be much higher. The force measured here is low, consistent with the 3.8 pN stall force measured in Feng.” (now reference 14). Not addressed.

Note for example motors run (creating force ramps) can stall (creating force plateaus) and can rupture. Sometimes they work individually, sometimes in small teams which would create plateaus at different higher force levels. The force magnitudes will also depend on the vector of the

underlying actin filament track relative to the pulling geometry of the sensor. Bonds under load will also show less Brownian motion and more stable sustained forces which appears to be the case here.

Figure 3A trajectories look like motors in teams. For 50FPS one or more at stall then a second one pulls a few times generating the peaks. For 10FPS a motor pulls ramping the force and then a second one pulls followed by rapid sequential rupture of both. Slower FPS would not see these individual trajectories but rather blend the whole force distribution. I really think the authors should consider this possibility. They have tried cytochalasin D. What about Blebbistatin? As mentioned in the first round review, stall force of individual motors attached to TCRs were measured with an optical trap to be 3.8 pN (ref 14) which is consistent with the nice data in this paper. Furthermore, after triggering TCRs stream towards the triggering site as seen in figure 5A of (ref 14) with anti-CD3 bead transport. Ref 14 also showed individual steps.

In summary, we really like the results here and believe there is a substantial amount of useful high quality data that would be good for the community to know about. There are a number of statements that seem beyond what can be interpreted from this work. We hope these can be cleaned up and tightened to what can be concluded based on this work. It is particularly important that the lipid attachment to the sensor be verified as able to sustain a load at or above the 42pN cited here. This could be measured or appropriately cited from other work using the same attachment chemistry. It is also important to make it clear that the earliest triggering event cannot be measured by this system, as some statements made in the manuscript are misleading on this front. It seems they are actually measuring motor transport of TCRs rather than triggering events.

Reviewer #1

Summary: *The revised manuscript by Göhring et al now includes a number of additional experiments that have further improved the manuscript. Since most of the reviewers' concerns have been adequately addressed (thanks for the very detailed rebuttal letter), I can fully support the publication of this interesting study.*

Answer: We thank the reviewer for this very positive assessment.

I only have a few suggestions:

Suggestion 1. *I would like to encourage the authors to include more of the supplementary data into the manuscript. Fig.3 and Fig. 4 are quite slim, while Supplementary Figs. 9 and 10 show very interesting data sets. Would it not be worth featuring these supplementary data in one of the main figures?*

Answer: We agree, and we have included previous **Supplementary Figure 9** as new **Figure 3** in the main text.

Suggestion 2. *The authors should check whether all references to the main figures are correct; the reference to Fig. 3B on page 14, for instance, is not fitting. (I think it should refer to Fig. 3C).*

Answer: We have carefully checked the references to the figures and corrected the wrong ones in the current version of the ms.

Reviewer #3

Summary: *The authors couple a fluorescence-based force sensor to both H57 and natural ligands that bind the T cell receptor. The ligands are displayed on lipid bilayers that are supported where they can sustain a force or fluid where they move in two dimensions. Two densities of sensors are used ensemble 50-100 molecules/ μm^2 and low/single molecule sensor of 0.01/ μm^2 interspersed within dark constructs at higher ligand density 50-100/ μm^2 such that activating conditions can be achieved (see page 10). These activating conditions are balanced with studies at low density “scanning”. There are lots of good results in the data and it would benefit the community to see this work.*

Answer: We thank the reviewer for this positive assessment.

Comment: *While I really like the measurements and design of the experiments, I think I have a different model than the authors for interpreting their measurements. In some places the authors overreach with respect to how T cells are activated. Below are a few of the particular statements and conclusions where I do not see support from the measurements that should be modified or addressed before publication. Other points for consideration and suggestions are included as well.*

Answer: We thank the reviewer for his/her constructive criticism. We believe we have addressed the points raised in the current version of the ms..

Comment: *In the abstract I do not understand the statement “recalibrate their significance in antigen recognition.” Much of this work addresses forces on TCRs with and without activation conditions being present, not the actual antigen recognition event / triggering / activation itself. What exactly is the conclusion here about antigen recognition? I don’t think the measurements are sensitive to these rare events. In their rebuttal to the first review the authors agreed that they do not have the temporal resolution to see early triggering events, however that is not expressed in the manuscript.*

Answer: We agree, and we have removed the statement from the abstract.

Comment: *Point to consider to avoid confusion in the field: The statement on page 7 second paragraph: “It is known that monomeric membrane-bound H57 activates T-cells with similar activation thresholds as agonistic...” should be qualified by stating that in free solution or under no force conditions H57 blocks activation. A reader not familiar with H57 might not understand H57Fab generally blocks activation except under bound conditions where force can be loaded.*

Answer: We thank the reviewer to maintain caution when working with artificial stimulating ligands. In the current version of the manuscript we support our statement with two references (Hellmeier et al. PNAS 2021 118:e2016857118; Lin et al. Biophys. J. 2020 118: 2879–2893): Both in 5c.c7 (Hellmeier) and AND T-cells (Lin), membrane-bound monovalent H57-based antibody fragments showed similar activation threshold compared to SLB-bound agonists.

We decided against adding further references considering the effect of H57 in solution on pMHC-triggered T-cell activation, since the experimental focus of our study is different. In addition, our own experiments (Huppa et al., Nature 2010 463:963-967) revealed unaffected calcium response and proliferation of 5c.c7 T-cells upon interaction with MCC-IE^k decorated SLBs in the presence or absence of H57 scF_v added in solution (see Figure 4c and supplementary figure 18). Furthermore, the effect of H57 Fab fragments on antigen-dependent proliferation of 2B4 T-cells was not discernable in a titration experiment performed in Johnson et al. PNAS 2000 97:10138-43; only at saturating concentrations a slight inhibition of proliferation was noticeable (see supplementary figure 6).

Comment: *In the introduction, p5 last paragraph it states “perpendicular tensile forces smaller than 2pN are already effective during the initiation of TCR signaling.” One cannot rule out short transients where higher force is exerted or rare events that are not seen, particularly since the number of unlabeled sensors are 3-4 fold higher than the labeled sensors. I think the 2pN the authors use arises*

from the average force distribution seen. (Short transients/spikes in figure 3A are seen going well above the average observed force.) In other studies at high ligand density external forces are not required. Yet it is possible that under these conditions force events occur within the dense interface arising from internal cell forces, the subject of the studies here. So I think it is a good point for the authors to address if possible as I feel it is an open important question. If there are potential transients this should be mentioned.

Answer: We understand and appreciate the reviewer's opinion, that single force spikes may be effective during the triggering process, which may be underrepresented in force distributions. To avoid too much speculation about this unresolved aspect, we rephrased the introduction section such that it does not rule out this possibility (line 95).

Comment: *A related point on Page 17 "(ii) Furthermore, our results render it unlikely that forces larger than 2 pN are imposed on the TCR are required to trigger the TCR..." Note that the only cases where the authors have triggered the TCR are for high ligand density, and T cells are known to trigger with as few as 1 or 2 molecules with force. This statement should be qualified for high ligand density.*

Answer: We added a clarifying apposition to this sentence (line 380).

Comment: *A point to consider: On page 15 top paragraph in the discussion of the activating conditions and scanning conditions for -IEk/MCC the sentence "The latter finding may result from early TCR-pMHC bond rupture..." It is likely that the lower force observed in activating compared to H57 also is due to early rupture.*

Answer: We agree with the reviewer's interpretation and changed the according sentences (line 334ff).

Comment: *The statement "Our observation of reduced pulling forces in the case of MFS-IEk/MCC would hence be in agreement with the presence of slip bonds formed between..." Note that as long as the bond lifetime is shorter than TCR-H57 (as is the case for both slip bonds and catch bonds in TCRs) one would expect to observe a reduced pulling force if the force is applied through some ramping mechanism (motor based or actin based transport) where the bond will let go before it gets to full force leading towards a reduced apparent force magnitude. I don't see how these results relate to catch or slip bond discussions.*

Discussion of slip bond and catch bond bottom of page 17/top of page 18. I think the catch bond only starts developing above 5 pN (See Das et al, PNAS 2015, Fig 4 and figs 2 and 3) I am not sure the speculation about the catch/slip bond is relevant here. Also see Hwang et al PNAS 2020 discussion of forces required to bring out catch bond organization.

Answer: In a hypothetical motor-based ramping mechanism, one may expect the motor-TCR connection to let go independently of the ligand. Hence, rupture before the maximum force magnitude, as reported by the MFS-H57 construct, would be indicative as slip bonds. We clarified this on line 404ff. However, we agree with the reviewer that the discussion on slip bonds versus catch bonds was too speculative and shortened it (line 411).

Point for consideration: *Discussion on page 18 regarding perpendicular pulling forces. As a cell lands on a SLB fluid surface, its cytoskeletal machinery is not necessarily organized relative to the plane of the coverglass. Thus with all of the invagination and local microstructure of the cell surface there is likely, especially early on, geometries of the bonding interface where TCRs can pull perpendicularly against the surface. As the cell flattens out however, the machinery gets organized largely more parallel to the surface where the sensors are no longer able to be pulled as opposed to dragged within the fluid SLB. I think this might be a better explanation for the elegant temporal progression results showing forces seen for landing cells.*

Answer: We thank the reviewer for this, again, very thoughtful suggestion, which could indeed explain alterations in force magnitude at early time-points under activating conditions. We added a sentence on line 266f.

Comment: *The introduction includes some good background on TCR forces through methods such as micropillars, deformable surfaces and even DNA based sensors but does not mention a number of single molecule studies where forces are directly measured on TCRs and motors linked to TCRs. It would be good to consider including some of these relevant works.*

Answer: We added a paragraph on motor-based transport of TCRs to the introduction section (line 53-57).

Comment: *P16 middle paragraph. The force required to pull the sensor is an estimate based on 6 lipids and 7 pN/lipid and a parallel anchor assumption. Based on the structure with 3 inactive sites it is not clear that this species will form parallel anchoring points in the lipid bilayer. I think it is necessary to verify the sensor can sustain higher force experimentally or cite a reference where the mSAv-3xHis₆ had been able to sustain loads well above the sensor range in a similar SLB. If the load is not shared evenly it could break below 42 pN. If the sensor breaks free from the lipid bilayer at ~7 pN this could be the reason no forces much above this value are observed.*

Answer: This point is well-taken. Recently, the lab of Erik Schäffer applied optical tweezers to characterize the maximum force that can be applied to a supported lipid bilayer system (Sudhakar et al., Nano Lett. 2019, 19:8877–8886). They used biotinylated lipids in a gel-phase supported lipid bilayer. The lipids were coupled to neutravidin, to which force was applied via an optical tweezer using beads linked to biotinylated DNA. Forces of up to 70pN could be applied to this system without pulling lipids out of the membrane. We have cited this study in the revised discussion (line 362f).

Point to consider: *Figure 2b plots track count vs. diffusion parameter. Some of the trajectories in supplementary Figure 8 look directed. Does the MSD slope show diffusion? For the mobile spots it would be interesting to know the velocity if they are directed. I am interested in this because my interpretation is that a number of these motions are actin-myosin based transport. How about plotting velocity as well as diffusion parameter?*

Answer: We have analyzed the trajectories by plotting the mean square displacement versus the time-lag (**Supplementary Figure S2A**, right panel, blue curve). However, there were no indications for deviations from Brownian motion. As the reviewer may be aware of, it is characteristic for pure Brownian trajectories to show segments indicative of both directed transport as well as confined diffusion.

For consideration: *On page 13 temporal analysis of scanning t cells is interesting. This might reflect reorganization of actin-myosin microstructure as the cell settles down.*

Answer: We agree with the reviewer, that this is very plausible. We have added a sentence in the discussion (line 424).

Point to consider: *On page 14 how long in time were the trajectories of the low FRET traces? Individual motors have a limited processivity. It is likely that these more stable higher frame rate trajectories are a single event and the longer time trajectories 71 seconds include many different motor/actin scenarios which would explain the increase in spread of force distribution observed.*

Answer: Again, we agree, this is a plausible scenario. We have mentioned in the revised version of our manuscript that the transient links may be formed via single motor protein molecules.

Comment: Discussion: Page 15 “a clutch-like coupling between cortical actin and microcluster-resident TCRs ...” Seems there is good evidence in this paper that it is motor.

Answer: Please refer to our response regarding the previous point.

Comment: *The statement “Interestingly, our time resolved experiments did not reveal... Such force ramps not only affect the mean but distribution of bond lifetimes...” From figure 3C, the loading rate appears to be about 2 pN/s. This is a very low loading rate case. That same figure shows trajectories that are flat over similar time windows. The data in figure 3C was taken from 7 single-molecule force ramps. What percentage of the time within these trajectories (duty cycle) show a force ramp? At least half of the 50 FPS trace appears flat and the second half of the 10 FPS trace in figure 3A. If much of the duty cycle is flat it can’t be claimed that there is always a force ramp. I suspect many of the motors are off or under stalled conditions.*

Answer: We agree with the reviewer, this would be a highly interesting question. In the current stage, however, we do not have sufficient statistics to determine duty cycles. The problem is not only to identify force ramps, but also to unambiguously identify regions without applied force. Some force transitions are difficult to detect due to noise in the signal, particularly the short ones. In addition, from the given force trajectories we cannot discriminate between rupture of the TCR-actin link and rupture of the TCR-MFS connection.

Comment: *Discussion page 19 about cytoskeletal drag, actin flow rates etc. It seems to me the results here are very consistent with motor-based transport of TCRs following activation. This was pointed out in the earlier review:*

From before: “I believe this current work may actually measuring force exerted on the TCR’s via actomyosin streaming after activation occurs somewhere else in time and space. It may be that the authors have nice evidence that streaming is single motor based. If it were coupled directly to an actin filament, force magnitudes through multiple motors might be much higher. The force measured here is low, consistent with the 3.8 pN stall force measured in Feng.” (now reference 14). Not addressed.

Note for example motors run (creating force ramps) can stall (creating force plateaus) and can rupture. Sometimes they work individually, sometimes in small teams which would create plateaus at different higher force levels. The force magnitudes will also depend on the vector of the underlying actin filament track relative to the pulling geometry of the sensor. Bonds under load will also show less Brownian motion and more stable sustained forces which appears to be the case here. Figure 3A trajectories look like motors in teams. For 50FPS one or more at stall then a second one pulls a few times generating the peaks. For 10FPS a motor pulls ramping the force and then a second one pulls followed by rapid sequential rupture of both. Slower FPS would not see these individual trajectories but rather blend the whole force distribution. I really think the authors should consider this possibility. They have tried cytochalasin D. What about Blebbistatin? As mentioned in the first round review, stall force of individual motors attached to TCRs were measured with an optical trap to be 3.8 pN (ref 14) which is consistent with the nice data in this paper. Furthermore, after triggering TCRs stream towards the triggering site as seen in figure 5A of (ref 14) with anti-CD3 bead transport. Ref 14 also showed individual steps.

Answer: We agree with the reviewer that it may very well be that individual motor proteins generate the observed force transitions. We have discussed this possibility in the revised version of our manuscript.

Summary: *In summary, we really like the results here and believe there is a substantial amount of useful high quality data that would be good for the community to know about. There are a number of statements that seem beyond what can be interpreted from this work. We hope these can be cleaned up and tightened to what can be concluded based on this work. It is particularly important that the lipid attachment to the sensor be verified as able to sustain a load at or above the 42pN cited here. This could be measured or appropriately cited from other work using the same attachment chemistry.*

It is also important to make it clear that the earliest triggering event cannot be measured by this system, as some statements made in the manuscript are misleading on this front. It seems they are actually measuring motor transport of TCRs rather than triggering events.

Answer: We thank the reviewer for this very positive assessment. In the revised version we have addressed the question concerning the load which can be sustained by the bilayer system: the cited optical tweezer experiments revealed maximum loads of 70pN even for single lipid molecules. This value is one order of magnitude higher than the observed pulling forces detected in our paper, and hence further warrant the design of our approach.

We believe that we have also clarified in our manuscript that the earliest TCR binding events, though included in our data, are likely masked by other binding events occurring at later stages. We rephrased the according sentence in the discussion section (line 384ff).

Reviewer #3 (Remarks to the Author):

The authors have appropriately and thoroughly addressed our concerns. This study is high quality of general interest and moves things forward. To the team there, impressive first rate work. We fully support publication.